# Hyperactivated glycolysis drives spatially patterned Kupffer cell depletion in MASLD

Jia He, Ran Li, Cheng Xie, Xiane Zhu, Keqin Wang, Zhao Shan*

Yunnan Key Laboratory of Cell Metabolism and Diseases, Center for Life Sciences, School of Life Sciences, Yunnan University, Kunming, China

## eLife Assessment

The authors aim to understand why Kupffer cells (KCs) die in metabolic-associated steatotic liver disease (MASLD). This is a **valuable** study using in vitro studies and an in vivo genetic mouse model, suggesting that increased glycolysis contributes to KC death in MASLD. The data presented are now **convincing** and adequately revised. This work will be of interest to researchers in the immunology and metabolism fields.

*For correspondence:
shanzhao@ynu.edu.cn

**Abstract** Metabolic dysfunction-associated steatotic liver disease (MASLD) progression is characterized by hepatic inflammation and cell death, yet the mechanisms underlying Kupffer cell (KC) loss remain poorly understood. Here, we sought to elucidate the metabolic basis of KC death during MASLD. Using metabolomics, immunostaining, and flow cytometry, we evaluated metabolic alterations and KC death throughout early MASLD progression. We found that KC death is an early hallmark of MASLD, exhibiting greater susceptibility and a spatial distribution consistent with KC zonation. Moreoever, KCs undergo progressive metabolic reprogramming toward enhanced glucose utilization during MASLD development, which is correlated with KC death. In combination with biochemical agonist, isotope tracing, and primary KC culture, we further demonstrated that augmented glycolytic metabolism directly drives KC death in vitro. Consistently, using *Chi3l1*-deficient mice, we further demonstrated that increased glucose utilization accelerates KC death in vivo. Together, these findings establish a causal link between glycolytic activation and KC loss during MASLD progression, highlighting glucose metabolic pathways as potential therapeutic targets to preserve KC homeostasis and mitigate MASLD.

## Introduction

Metabolic dysfunction-associated steatotic liver disease (MASLD) emerges as a prevalent liver condition in the Western world, affecting roughly one-third of the population (*Hardy et al., 2016*; *Younossi et al., 2018*; *Younossi, 2019*). Its incidence continues to climb due to its strong association with obesity, type 2 diabetes, and the metabolic syndrome (*Eslam et al., 2020a*; *Rinella and Sanyal, 2016*; *Sheka et al., 2020*). MASLD includes a spectrum of liver disorders, from metabolic dysfunction-associated fatty liver, often referred to as steatosis, to metabolic dysfunction-associated steatohepatitis (MASH) (*Eslam et al., 2020b*). MASH, characterized by steatosis, inflammation, ballooning injury, and varying degrees of fibrosis, marks the initial critical stage of MASLD (*Friedman et al., 2018*). Kupffer cells (KCs) are liver-resident macrophages located in the hepatic sinusoids (*Gomez Perdiguero et al., 2015*; *Kazankov et al., 2019*). Derived from the embryonic yolk sac, KC can self-renew through proliferation during adult homeostasis (*Hashimoto et al., 2013*). Studies suggest that

KC contribute to triglyceride (TG) storage during MASH progression, as demonstrated by depleting CD207[+] KC in CD207-DTR mice using diphtheria toxin or by using CD207[ΔBcl2l1] mice to stimulate a low embryo-derived KC (EmKC) status (*Tran et al., 2020*). Upon KC death, monocyte-derived macrophages (MoMFs) gradually seed the KC pool and eventually replace the deceased KC (*Scott et al., 2016*). These MoMFs tend to be more inflammatory than EmKC, altering the liver's response in MASH, eventually limiting TG storage and contributing to liver fibrosis (*Daemen et al., 2021*; *Tran et al., 2020*). Researchers, including our team, have observed a gradual decline in KCs during MASLD development (*Daemen et al., 2021*; *He et al., 2025*; *Remmerie et al., 2020*; *Tacke, 2017*; *Tran et al., 2020*). However, little is known regarding the dynamic loss of KCsand metabolic changes behind KC death during MASLD.

Emerging evidence highlights a fundamental role of glucose metabolism in regulating macrophage function, polarization, and survival (*Du et al., 2019*; *Faas et al., 2021*; *Ma et al., 2020*; *Woods et al., 2022*). Metabolic reprogramming is a well-established hallmark of macrophage activation and functional adaptation across diverse contexts (*Murray et al., 2014*). Within the unique metabolic milieu of the MASLD liver, characterized by lipotoxicity, insulin resistance, and disrupted nutrient fluxes, it is plausible that KC metabolism is profoundly altered. However, whether these metabolic perturbations directly contribute to the observed loss of KCs, and through which specific metabolic pathways, remains unclear. Therefore, this study aims to investigate the role of glucose metabolism in governing KC susceptibility to death during MASLD progression. We sought to define the glucose metabolic alterations that occur in KCs as MASLD develops and to establish a causal link between metabolic reprogramming and KC loss. Elucidating these mechanisms is essential for advancing our understanding of MASLD pathogenesis and for identifying potential therapeutic targets to preserve KC homeostasis and mitigate disease progression.

## Results

### The death of KCs is a pathological characteristic during MASLD

To systematically investigate KC death, we established an MASLD mouse model by feeding wild-type (WT) C57Bl/6J mice a high-fat high-cholesterol diet (HFHC) (*Figure 1—figure supplement 1A*). We then performed hematoxylin and eosin (H&E), Oil Red O, and Sirius Red staining to assess immune cell infiltration, fat accumulation, and liver fibrosis in mice fed HFHC for 0, 4, or 16 weeks. After 4 weeks of HFHC feeding, we observed a slight increase in lipid droplets, with no apparent signs of liver inflammation or fibrosis in hepatocytes (*Figure 1—figure supplement 1A*). By 16 weeks of HFHC feeding, both lipid droplets and immune cell infiltration had increased significantly, though liver fibrosis—as indicated by Sirius Red staining, which labels collagen deposition—was still not induced, and MASLD activity score remained below 4 (*Figure 1—figure supplement 1A*). Moreover, the body weight of mice fed HFHC gradually increased over the feeding period (*Figure 1—figure supplement 1B*). Analysis of serum alanine aminotransferase (ALT), aspartate aminotransferase (AST), serum and liver cholesterol, or liver TG levels revealed significant increases compared to mice fed a normal chow diet (NCD), except for serum TG, which was similar between NCD and HFHC-fed mice (*Figure 1—figure supplement 1C*). These findings collectively indicate the successful establishment of an early MASLD mouse model, without fibrosis development.

Subsequently, we investigated KC death by labeling KCs with antibodies targeting T cell immunoglobulin mucin protein 4 (TIM4) (*Scott et al., 2018*; *Tran et al., 2020*) and employing TdT-mediated dUTP Nick-End Labeling (TUNEL) to detect dead cells. Compared to baseline (0 week, prior to HFHC), KCs are significantly reduced with MASLD progression. Approximately 25% and nearly 60% of KCs underwent cell death by 4 and 16 weeks post-HFHC feeding, respectively (*Figure 1A and B*). To further validate KC death during MASLD progression, we performed co-staining of the apoptotic marker Cleaved caspase 3 (Cl-Casp3) with C-type lectin domain family 4 (Clec4f), another marker specific for KCs (*Scott et al., 2016*). Although the ratio of Cl-Casp3[+]/Clec4f[+] was relatively lower, its occurrence and gradual increase followed a trend similar to that of the TUNEL staining (*Figure 1—figure supplement 2A*). Considering the potential recruitment of MoMFs into the liver, some of which may acquire KC features, including TIM4 and Clec4f expression (*Daemen et al., 2021*; *Tran et al., 2020*), we performed flow cytometry analysis of liver nonparenchymal cells (NPCs) to distinguish resident KCs from monocyte-derived populations. Resident KCs were identified as CD45[+] F4/80[hi]

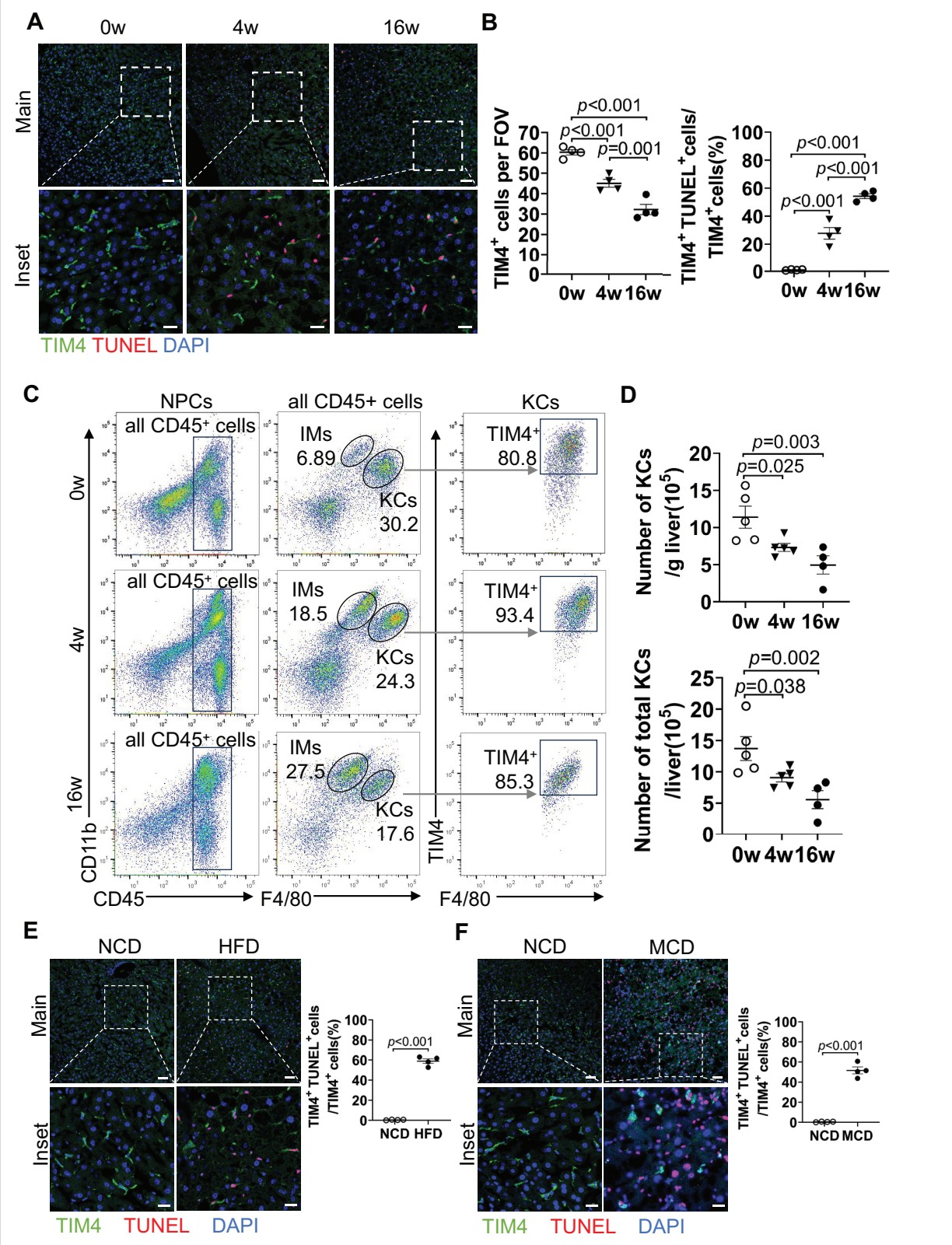

**Figure 1.** Kupffer cell (KC) death is a characteristic feature of metabolic dysfunction-associated steatotic liver disease (MASLD) progression. (**A–D**) Male wild-type C57BL/6J mice were fed a high-fat high-cholesterol diet (HFHC) for 0, 4, or 16 weeks. (**A**) KC death was assessed by immunostaining of liver sections for TIM4 (KC marker, red), TdT-mediated dUTP Nick-End Labeling (TUNEL) (green), and DAPI (nuclei, blue). Scale bar: 50 μm (main panels) and 20 μm (inset). (**B**) KC death was quantified. n=4 mice/group. (**C**) Flow cytometry analysis of KCs (CD45[+] F4/80[hi] CD11b[low] TIM4[+]) and infiltrating

*Figure 1 continued on next page*

*Figure 1 continued*

macrophages (IMs) (CD45[+] F4/80[low]CD11b[hi] TIM4[-]) among isolated nonparenchymal cells (NPCs). (**D**) KC counts were quantified. n=4–5 mice/group. (**E–F**) Male wild-type C57BL/6J mice were fed either: (**E**) normal chow diet (NCD) or high-fat diet (HFD) for 20 weeks, or (**F**) NCD or methionine-choline-deficient diet (MCD) for 6 weeks. KC death was assessed by immunostaining of liver sections for TIM4 (green), TUNEL (red), and DAPI (nuclei, blue). Scale bar: 50 μm (main panels) and 20 μm (inset). KC death was quantified. n=4 mice/group. Representative images are shown in A, C, E, F. One-way ANOVA (**B, D**). Unpaired Student's t-test (**E, F**). p-Value as indicated.

The online version of this article includes the following source data and figure supplement(s) for figure 1:

**Source data 1.** Numerical data of *Figure 1B–D and E–F*.

**Figure supplement 1.** The generation of high-fat high-cholesterol diet (HFHC)-induced metabolic dysfunction-associated steatotic liver disease (MASLD) mouse model.

**Figure supplement 1—source data 1.** Numerical data of *Figure 1—figure supplement 1A–C*.

**Figure supplement 2.** Examination of Kupffer cell (KC) death and monocyte-derived macrophages (MoMFs) recruitment in high-fat high-cholesterol diet (HFHC) mice.

**Figure supplement 2—source data 1.** Numerical data of *Figure 1—figure supplement 2A and C–D*.

CD11b[low] TIM4[hi] cells, MoKCs as CD45[+] F4/80[hi] CD11b[low] TIM4[low] cells, and infiltrating macrophages (IMs) as CD45[+] F4/80[+] CD11b[hi] TIM4[-] cells (***Figure 1C***). Our analysis showed that all TIM4[+] KCs were of EmKCs, with no detectable MoKCs at any time point examined (***Figure 1C***). Consistent with the immunostaining results, the number of resident KCs decreased as early as 4 weeks and continued to decline through 16 weeks of HFHC feeding (***Figure 1C and D***). In contrast, IMs progressively accumulated during HFHC feeding (***Figure 1C and D***). To further refine the identification of MoMFs, we exclude CD45[+] F4/80[+] CD11b[hi] TIM4[-] Ly6G[+] neutrophils from the IM gate. This refined analysis yielded similar results, confirming a gradual increase in MoMFs over the course of HFHC feeding (***Figure 1—figure supplement 2B and C***).

To exclude TUNEL false positivity associated with proliferation, we evaluated Ki67 expression in dying KCs (TUNEL[+]TIM4[+]). We observed that proliferating KCs (Ki67[+]TIM4[+]) were infrequent throughout HFHC feeding, and less than 15% of TUNEL[+]TIM4[+] KCs co-expressed Ki67. This indicates that the majority of TUNEL[+] KCs represent true cell death events rather than proliferation-related artifacts (***Figure 1—figure supplement 2D***). To examine the broader occurrence of this phenomenon in MASLD, we further assessed KC death in several other dietary mouse models, including mice fed a high-fat diet (HFD) for 20 weeks (***Zhang et al., 2022***) and a methionine/choline-deficient diet (MCD) for 6 weeks (***Tran et al., 2020***). Co-staining of TIM4 and TUNEL in liver sections revealed a notable increase in KC death in both the HFD and MCD groups compared to the control group fed an NCD (***Figure 1E and F***). These findings collectively confirm that progressive KC death is a pathological hallmark of MASLD observed across various dietary-induced models.

## KCs exhibit early and zone-specific susceptibility to death during MASLD progression

To compare relative susceptibility of different hepatic cell populations to MASLD-induced death, we performed TUNEL co-staining with lineage markers: HNF4α (hepatocytes), Desmin (hepatic stellate cells [HSCs]), and Iba1 (hepatic macrophages, including both KCs and MoMFs) (***Figure 2A–C***). Hepatocyte death showed minimal change during the initial 4 weeks but increased modestly by 16 weeks (***Figure 2A***). In contrast, HSC and hepatic macrophage mortality rose progressively throughout MASLD progression, with significant increases detectable as early as 4 weeks (***Figure 2B and C***). Notably, the finding that 60% of TIM4[+] KCs are TUNEL[+], compared to 30% of total Iba1[+] cells, further supports that resident KCs undergo death more readily than the broader macrophage pool, which includes MoMFs.

We next determined whether KC death displays spatial zonation within the liver lobule. Based on the lobular architecture—comprising periportal (PP, zone 1), midzonal (Mid, zone 2), and pericentral (PC, zone 3) regions (***Ben-Moshe et al., 2022***)—we performed co-staining for TIM4, TUNEL, and glutamine synthetase (*GS; a central vein marker*) (***Figure 2D***). Spatial analysis revealed an early zonation bias in KC death. At 8 weeks of HFHC feeding, the proportion of TIM4[+]TUNEL[+] cells among total TIM4[+] KCs was significantly higher in PV regions compared to CV regions (p=0.041), indicating increased susceptibility of periportal KCs (***Figure 2D***). However, by 16 weeks, although absolute numbers of apoptotic KCs differed across zones, the proportional death rate became comparable

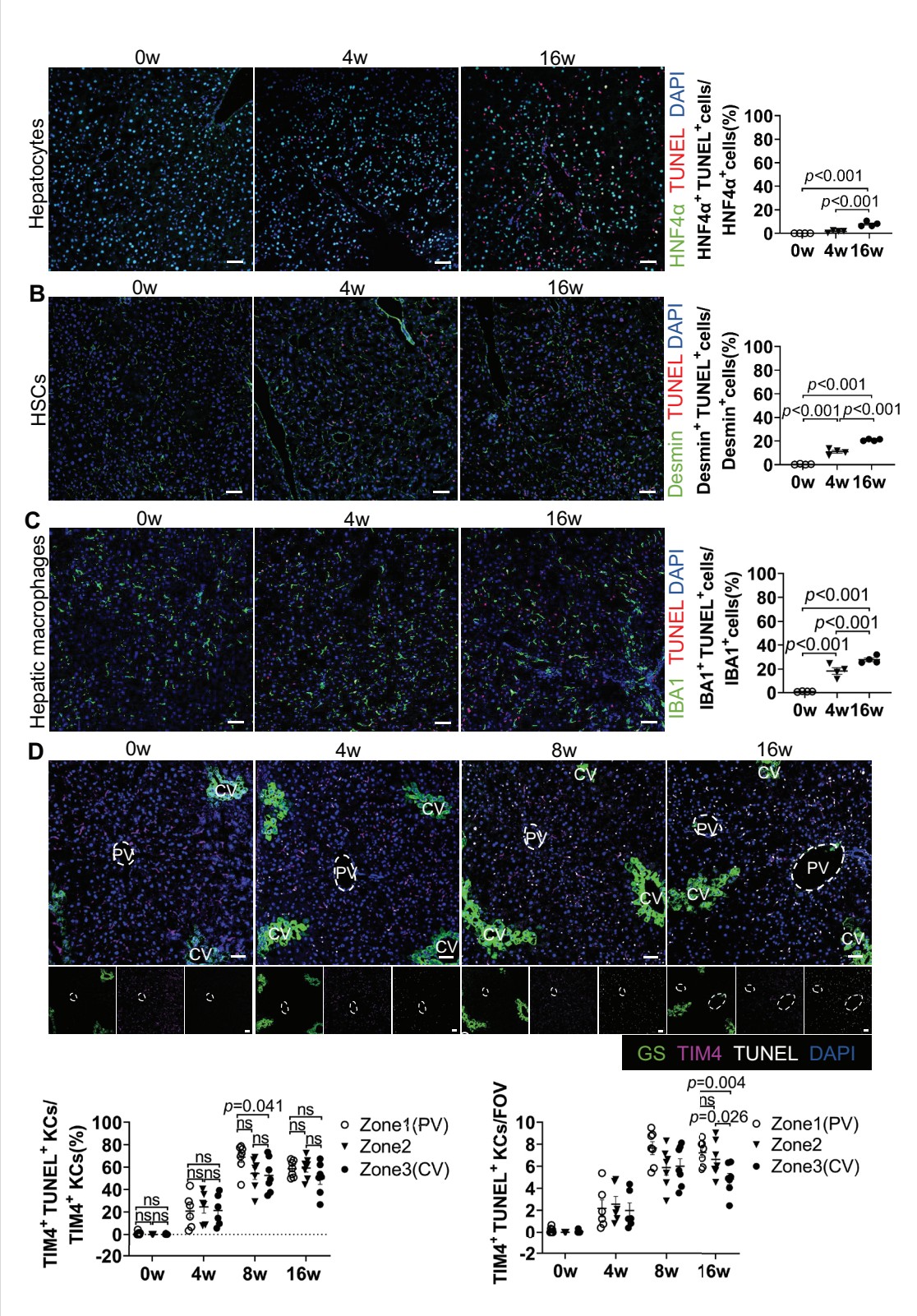

**Figure 2.** Kupffer cells (KCs) exhibit early and zone-specific susceptibility to death during metabolic dysfunction-associated steatotic liver disease (MASLD) progression. (A–E) Male wild-type C57BL/6J mice were fed a high-fat high-cholesterol diet (HFHC) for 0, 4, or 16 weeks. (A–C) Hepatic cell death was assessed by co-staining TdT-mediated dUTP Nick-End Labeling (TUNEL) with: (A) HNF4α (hepatocytes), (B) Desmin (hepatic stellate cells [HSCs]), (C) Iba1 (hepatic macrophages), and DAPI (nuclei, blue). Scale bars: 50 μm (main panels). Hepatic cell death was quantified (n=4 mice/

*Figure 2 continued on next page*

*Figure 2 continued*

group). (**D**) Zonal distribution of KC death was evaluated by co-staining TIM4 (KCs), TUNEL, glutamine synthetase (GS, central vein marker), and DAPI (nuclei, blue). Scale bars: 50 µm. Zonal distribution of KC death was quantified (n=6–7 mice/group). FOV: field of view. PV: portal vein. CV: central vein. Representative images are shown in A–D. One-way ANOVA (**A–D**). p-Value as indicated.

The online version of this article includes the following source data for figure 2:

**Source data 1.** Numerical data of *Figure 2A–D*.

(*Figure 2D*), suggesting that prolonged HFHC feeding leads to widespread KC loss that attenuates early zonation bias.

## KCs exhibit metabolic reprogramming with increased glycolysis during early MASLD

To define the glucose metabolic alterations that occur in KCs during MASLD development, we isolated KCs from WT mice at various time points of HFHC feeding (0, 4, 8, and 16 weeks). The purity of isolated KCs was confirmed by TIM4 immunofluorescence staining, with TIM4$^+$ cells exceeding 90% (*Figure 3—figure supplement 1A*). We then conducted qRT-PCR to assess the mRNA expression levels of key enzymes involved in glycolysis (*Slc2a1, Hk3, Pfkfb3, Pkm*), the pentose phosphate pathway (PPP) (*G6pd, 6pdg*), glycogenolysis (*Pygl*), and glycogenesis (*Gys1, Ugp2*). Our data revealed that mRNA expression of rate-limiting enzymes for fast glucose metabolism, such as glycolysis and PPP, was significantly increased as early as 8 weeks after initiating the HFHC (*Figure 3—figure supplement 1B*). While glycolytic enzyme expression remained elevated, PPP enzyme expression began to decline by 16 weeks (*Figure 3—figure supplement 1B*). In contrast, enzymes linked to slower glucose metabolism, such as oxidative phosphorylation (*Idh1, Ogdh*), did not exhibit significant changes (*Figure 3—figure supplement 1B*). Furthermore, mRNA expression of glycogenesis and glycogenolysis rate-limiting enzymes started to decrease at 8 and 16 weeks, respectively, suggesting that glucose uptake becomes the primary source of KC glucose metabolism during this period (*Figure 3—figure supplement 1B*). Additionally, we investigated mRNA expression of rate-limiting enzymes involved in β-oxidation (*Acadm, Hadh*) but found no significant differences (*Figure 3—figure supplement 1B*). These data suggest a time-dependent metabolic inflexibility, where impaired glycogen handling and mitochondrial inertia drive a sustained glycolytic dependence in KCs, which may exacerbate their vulnerability (particularly in periportal zones, where glycogen storage is predominant; *Okada et al., 2025*).

To validate KC-specific metabolic alterations in MASLD, we performed metabolomic analysis on primary KCs from WT mice fed an HFHC for 0, 4, or 8 weeks (*Figure 3A*). Given that KC death peaked at 8 weeks in our model (*Figure 2D*), we focused on these early time points to capture initial metabolic shifts. Principal component analysis (PCA) of KC metabolites revealed distinct, diet duration-dependent clustering (*Figure 3B*). Kyoto Encyclopedia of Genes and Genomes (KEGG) pathway enrichment analysis identified glucose metabolism pathways—including glycolysis and the PPP—as the most significantly upregulated during MASLD progression (*Figure 3C and D*). This glycolytic activation was further corroborated by time-dependent accumulation of key intermediates, including glucose, phosphatidylethylamine (PEA), phosphoenolpyruvate (PEP), fructose-1,6-bisphosphate (FBP), and lactate (LA) in heatmap analysis (*Figure 3E*). Critically, we observed progressive increases in pro-apoptotic metabolites generated through these glucose metabolism pathways: redox disruptors (GSSG, FAD), mitochondrial toxins (methylmalonic acid), and apoptosis mediators (Hcy)—all exhibiting temporal coupling with glycolytic intermediates (*Figure 3F*). This demonstrates that KCs undergo rapid glycolytic reprogramming during early MASLD pathogenesis, which actively generates cytotoxic effectors coinciding with their peak vulnerability.

## Excessive glucose metabolic activity contributes to KC death

To investigate the direct role of glucose metabolism in KC death, isolated KCs were subjected to in vitro metabolic perturbations. First, KCs were treated with a combination of high glucose and palmitic acid (PA) to model MASLD pathology. Physiological glucose (5.5 mM) combined with PA (800 µM) increased KC death by approximately 10%. Strikingly, glucose at a concentration mimicking HFHC feeding (10 mM) combined with PA (800 µM) increased KC death by approximately 27%. This

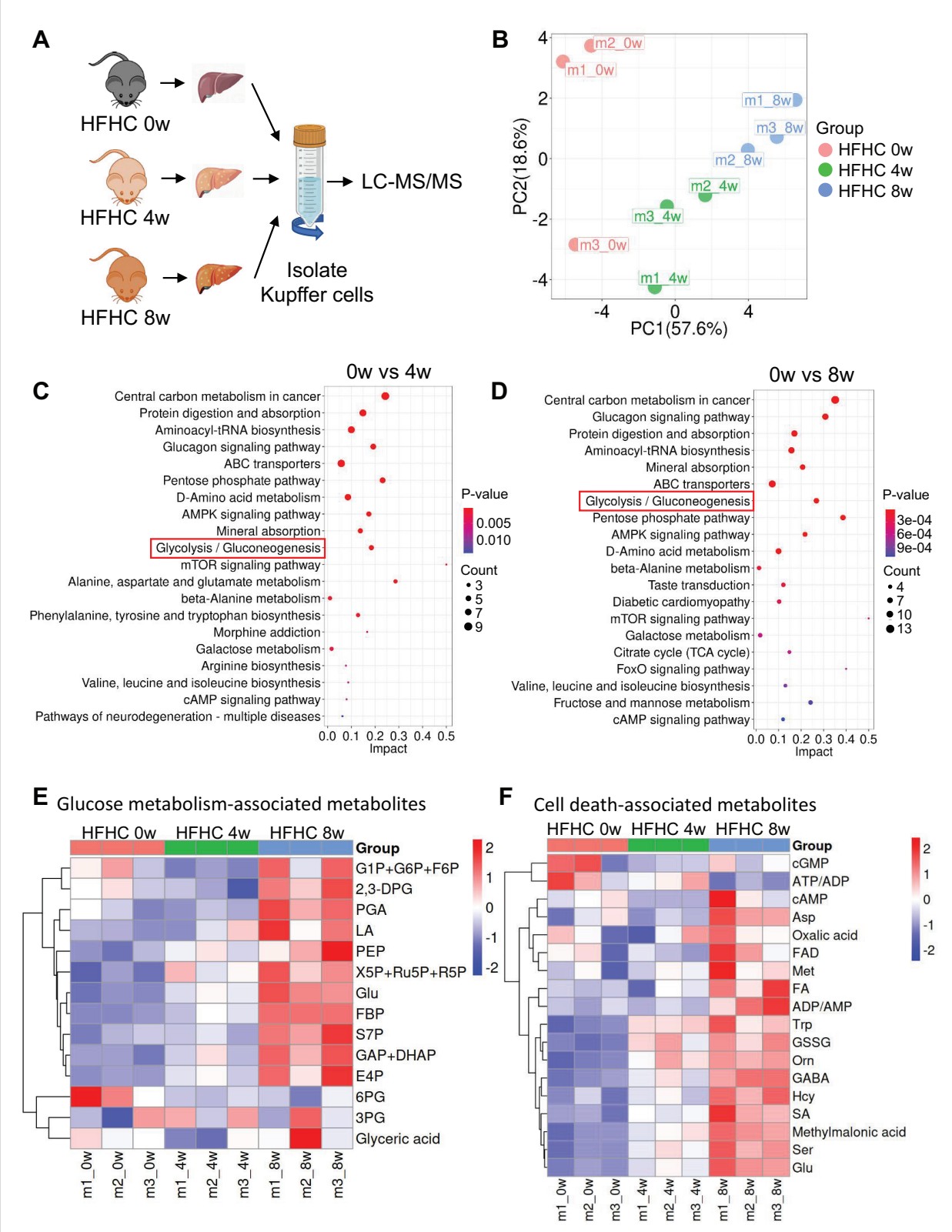

**Figure 3.** Kupffer cells (KCs) exhibit metabolic reprogramming with increased glycolysis during early metabolic dysfunction-associated steatotic liver disease (MASLD). (**A**) Experimental design for metabolomic analysis of KCs isolated from male wild-type mice fed a high-fat high-cholesterol diet (HFHC) for 0, 4, or 8 weeks. n=3 mice/group. (**B**) Principal component analysis (PCA) of enriched metabolites in KCs across different dietary durations. (**C–D**) Kyoto Encyclopedia of Genes and Genomes (KEGG) pathway enrichment analysis of metabolic pathways upregulated in KCs at 4 weeks (**C**) or

*Figure 3 continued on next page*

*Figure 3 continued*

8 weeks (**D**). The glucose metabolism pathway is highlighted by red rectangles. (**E**) Heatmap depicting significantly altered metabolites involved in glucose metabolism pathways in KCs across different dietary durations. (**F**) Heatmap depicting significantly altered metabolites involved in cell death in KCs across different dietary durations.

The online version of this article includes the following source data and figure supplement(s) for figure 3:

**Source data 1.** Numerical data of *Figure 3B–F*.

**Figure supplement 1.** Dynamic changes in mRNA expression levels of rate-limiting enzyme genes involved in glucose metabolism.

**Figure supplement 1—source data 1.** Numerical data of *Figure 3—figure supplement 1A–B*.

was evidenced by increased Cl-Casp3 staining (*Figure 4A*) and elevated Cl-Casp3 protein levels via western blot (*Figure 4B*).

Second, we treated KCs with the pyruvate dehydrogenase kinase (PDK1) activator PS48, which directly stimulates glycolysis. The results indicated a markedly increased KC death compared to blank or DMSO vehicle controls (*Figure 4C and D*). Finally, oligomycin (an ATP synthase inhibitor) was used to force glycolytic reliance by blocking mitochondrial ATP production. Oligomycin treatment similarly induced significant KC death (*Figure 4—figure supplement 1A and B*), phenocopying the effects of direct glycolytic activation. KCs are difficult to culture and prone to death in vitro. To assess the health status of KCs in our metabolic perturbation assays, we also examined KC viability using Calcein-AM staining, a fluorescent dye that labels metabolically active cells. Image analysis confirmed that the KCs were healthy and alive under all tested conditions (*Figure 4—figure supplement 1C*), indicating that our isolated KCs were suitable for in vitro experiments.

To determine whether enhanced glycolytic activity contributes to KC death in vivo during MASLD, we pharmacologically inhibited glycolysis using 2-deoxy-D-glucose (2-DG). 2-DG functions as a glucose analog that enters cells but cannot be fully metabolized, thereby inhibiting glycolysis through blockade of hexokinase and phosphoglucose isomerase (*Figure 4E*). WT C57BL/6J mice were fed an HFHC for 5 weeks to induce early-stage MASLD. To inhibit KC glycolysis in vivo, mice received intraperitoneal injections of either vehicle or 2-DG (50 mg/kg) every other day beginning at week 3 of HFHC feeding. The relatively short duration of HFHC exposure (5 weeks) combined with low-dose 2-DG treatment was designed to minimize potential confounding effects of systemic glycolytic inhibition on hepatocytes. At the end of the 5-week feeding period, KC survival was assessed by co-immunostaining liver sections for the KC marker TIM4 and TUNEL to detect apoptotic cells (*Figure 4F*). Quantitative analysis demonstrated that 2-DG-treated mice exhibited a significantly lower proportion of TUNEL[+] KCs compared with vehicle-treated controls (*Figure 4G*). These results indicate that pharmacological inhibition of glycolysis protects KCs from cell death during MASLD. Together with our in vitro data showing that glycolytic activation promotes KC death.

## Enhanced glycolytic flux in *Chi3l1⁻ᐟ⁻* macrophages

To investigate whether hyperactivated glycolysis specifically in KCs drives their death *in vivo*, we focused on Chitinase 3-like 1 (Chi3l1; gene *Chi3l1*)—a known inhibitor of glucose uptake (*He et al., 2025*). Since Chi3l1 suppresses glucose uptake by KCs, KCs in *Chi3l1⁻ᐟ⁻* mice are expected to maintain a state of hyperactivated glycolysis. To confirm this, we first performed a glucose metabolic flux assay in KCs. However, due to the limited availability of primary KCs, we used bone marrow-derived macrophages (BMDMs) to dissect the Chi3l1-dependent metabolic mechanisms. Uniformly labeled [U-$^{13}$C]glucose tracer analysis (*Figure 5A*) revealed genotype-specific metabolic reprogramming: PCA showed distinct clustering between *Chi3l1⁻ᐟ⁻* and WT BMDMs (*Figure 5B*), and heatmaps demonstrated pronounced accumulation of glycolytic intermediates (*Figure 5C*). Metabolic flux quantification confirmed significantly elevated U-$^{13}$C enrichment in glycolytic metabolites—including glucose (Glc), fructose-6-phosphate (F6P), 3-phosphoglycerate (3PGA), 2-phosphoglycerate (2PGA), PEP, pyruvate (PA), and LA—whereas glucose-6-phosphate (G6P), FBP, and PPP intermediates (ribulose-5-phosphate [Ru5P], ribose-5-phosphate [R5P], sedoheptulose-7-phosphate [S7P]) remained unchanged. Dihydroxyacetone phosphate (DHAP) was the only PPP-linked metabolite showing increased enrichment (*Figure 5D*), indicating that *Chi3l1* deletion selectively hyperactivates glycolysis without engaging the PPP. Furthermore, extracellular acidification rate (ECAR) analysis also showed significantly elevated glycolytic capacity in *Chi3l1⁻ᐟ⁻* BMDMs (*Figure 5E and F*).

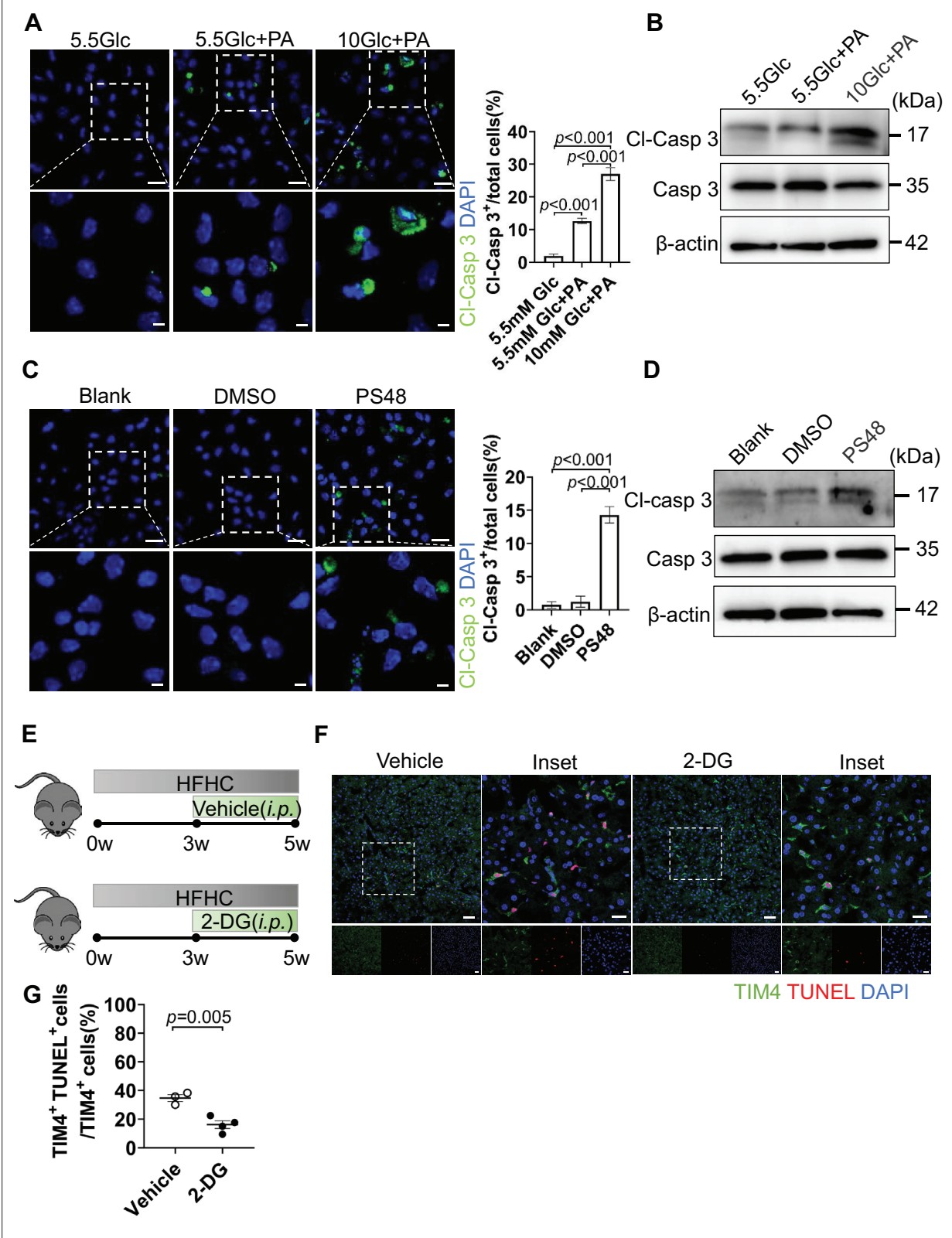

**Figure 4.** Excessive glucose metabolic activity contributes to Kupffer cell (KC) death. (**A–B**) Isolated KCs were treated for 24 hr with: 5.5 mM glucose+isopropanol (control), 5.5 mM glucose+800 μM palmitic acid (PA), 10 mM glucose+800 μM PA. Cell viability was assessed by Cleaved caspase 3 (Cl-Casp3) staining (Cl-Casp3+ cells = dead). Scale bars: 20 μm (main panels), 5 μm (insets). Cl-Casp3 was detected by western blot. (**C–D**) Isolated KCs were treated for 24 hr with: blank (no treatment), DMSO (vehicle control), 20 μM PS48 (PDK1 activator). Scale bars: 20 μm (main panels), 5 μm (insets).

*Figure 4 continued on next page*

*Figure 4 continued*

Cell death was analyzed as above. (**E**) Experimental design. Male wild-type (WT) mice were fed a high-fat high-cholesterol diet (HFHC) for 5 weeks. From the third week onward, mice received intraperitoneal injections of either vehicle or 2-DG (50 mg/kg) every other day (n=3–4 mice/group). (**F–G**) Effects of glycolysis inhibition on KC death after 5 weeks of HFHC feeding. (**F**) Representative images of liver sections co-stained with TdT-mediated dUTP Nick-End Labeling (TUNEL) and the KC marker TIM4. (**G**) Quantification of TUNEL⁺ KCs. Data are presented as mean ± SEM. Statistical analysis was performed using one-way ANOVA (**A, C**) and an unpaired Student's t-test (**G**); p-values are indicated.

The online version of this article includes the following source data and figure supplement(s) for figure 4:

**Source data 1.** Numerical data of *Figure 4A, C and G*.

**Source data 2.** PDF file containing original western blots for *Figure 4B and D*, indicating the relevant bands and treatments.

**Source data 3.** Original files for western blot analysis displayed in *Figure 4B and D*.

**Figure supplement 1.** Excessive glucose metabolic activity contributes to Kupffer cell (KC) death.

**Figure supplement 1—source data 1.** Numerical data of *Figure 4—figure supplement 1B*.

**Figure supplement 1—source data 2.** PDF file containing original western blots for *Figure 4—figure supplement 1C*, indicating the relevant bands and treatments.

**Figure supplement 1—source data 3.** Original files for western blot analysis displayed in *Figure 4—figure supplement 1C*.

Increased glycolysis is often associated with proinflammatory polarization of macrophages (*Tacke, 2017*). To assess the effect of *Chi3l1* loss on KC polarization, we further isolated KCs from WT and *Chi3l1⁻/⁻* mice fed an HFHC at 0, 8, and 16 weeks, and measured the mRNA expression of proinflammatory markers (*Nos2, Cxcl9, Ciita, Cd86, Ccl3,* and *Ccl5*) and anti-inflammatory markers (*Chil3, Retnla, Arg1,* and *Mrc1*). In WT KCs, the mRNA levels of proinflammatory markers increased with MASLD progression: *Nos2, Cxcl9, CIITA,* and *Ccl3* rose significantly, while *CD86* and *Ccl5* showed an upward trend (*Figure 5—figure supplement 1A*). In contrast, *Chi3l1⁻/⁻* KCs exhibited higher baseline expression of proinflammatory genes, with *Nos2* and *Ccl5* significantly elevated and *Cxcl9* and *CD86* showing an increasing tendency compared to WT. As MASLD developed, proinflammatory gene expression increased further in *Chi3l1⁻/⁻* KCs. No significant differences in anti-inflammatory marker expression were observed between WT and *Chi3l1⁻/⁻* KCs at baseline. However, with MASLD progression, *Chi3l1⁻/⁻* KCs displayed a rising trend in the expression of anti-inflammatory markers, including *Chil3, Arg1,* and *Mrc1* (*Figure 5—figure supplement 1A*). Together, these data suggest that Chi3l1 deficiency drives KCs toward a partially proinflammatory phenotype.

Since Chi3l1 primarily functions in its secretory form as a macrophage glucose uptake inhibitor (*He et al., 2025*), we also added recombinant Chi3l1 (rChi3l1) supplementation group. rChi3l1 supplementation reversed the hyper-glycolytic flux, restoring it to levels comparable to those in WT macrophages (*Figure 5—figure supplement 2A–D*). This was accompanied by reduced mass isotopologue distributions of key glycolytic intermediates (Glc, F6P, FBP, 3PGA, 2PGA, PEP, PA, LA, G6P) (*Figure 5—figure supplement 2B and C*). In contrast, PPP intermediates (Ru5P, R5P, S7P, DHAP) were unaffected (*Figure 5—figure supplement 2B and C*), confirming the specificity of the effect on glycolysis. Functionally, rChi3l1 significantly lowered lactate dehydrogenase (LDH) activity in high glucose-treated BMDMs, further corroborating the attenuation of glycolytic output (*Figure 5—figure supplement 2D*). We acknowledge that detailed metabolic flux analysis was performed in BMDMs rather than primary KCs due to the technical challenge of obtaining sufficient viable cells for such assays. While ontogenically distinct from KCs, BMDMs are a widely accepted model for mechanistic metabolic dissection. Importantly, the hyperglycolytic phenotype observed in *Chi3l1⁻/⁻* BMDMs was consistent with our in vivo KC data: Chi3l1 deficiency drove proinflammatory polarization. Thus, despite this limitation, the concordance between our in vitro mechanistic findings and in vivo phenotypic data provides compelling evidence that *Chi3l1⁻/⁻* mice serve as a mechanistic model to investigate glycolysis-driven KC death during MASLD. Collectively, these data establish that Chi3l1 reduces glycolytic flux—without affecting PPP activity—in macrophages, thereby providing a mechanistic model to investigate glycolysis-driven KC death.

## Enhanced glycolysis drives KC death in MASLD

We therefore employed *Chi3l1⁻/⁻* mice as a mechanistic model to investigate glycolysis-driven KC death. First, we isolated primary KCs from WT and *Chi3l1⁻/⁻* mice and treated them with PA or Iso, or left untreated (blank). *Chi3l1⁻/⁻* KCs exhibited significantly increased susceptibility to PA-induced

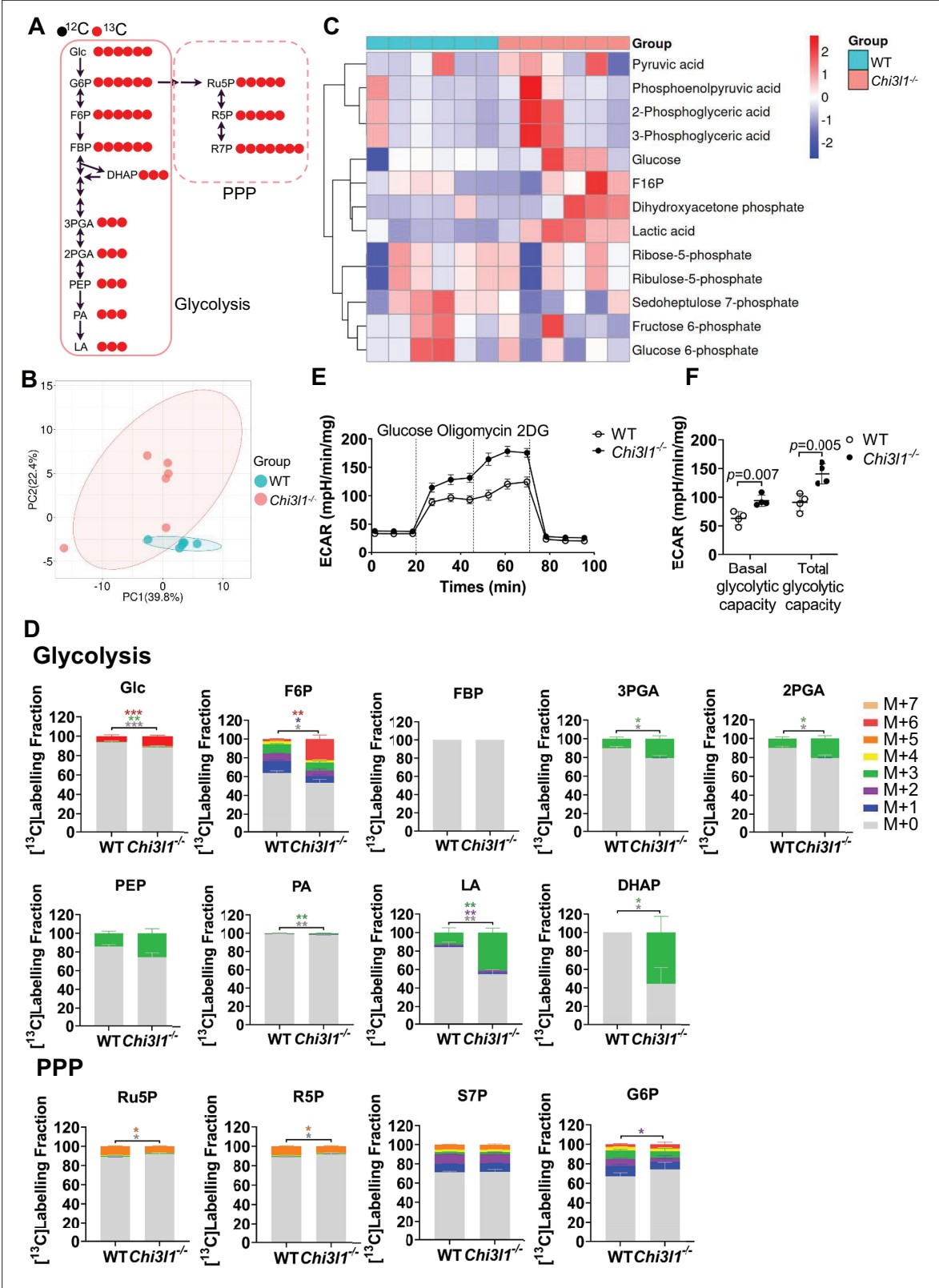

**Figure 5.** Enhanced glycolytic flux in *Chi3l1⁻/⁻* macrophages. (**A**) Schematic diagram depicting the fate of glucose-derived ribose carbons in wild-type (WT) mouse macrophages. (**B**) Principal component analysis (PCA) of metabolites in WT and *Chi3l1⁻/⁻* bone marrow-derived macrophages (BMDMs) cultured with [U-¹³C]glucose. (**C**) Heatmap depicting significantly altered glycolysis and pentose phosphate (PPP) metabolites in WT and *Chi3l1⁻/⁻* BMDMs. (**D**) Glucose metabolic flux analysis in WT and *Chi3l1⁻/⁻* BMDMs cultured with [U-¹³C]glucose showing mass isotopologue distributions of:

*Figure 5 continued on next page*

*Figure 5 continued*

glycolytic intermediates (Glc, F6P, FBP, 3PGA, 2PGA, PEP, PA, LA, G6P) and PPP intermediates (Ru5P, R5P, S7P, DHAP). Data represent n=6 biological replicates/group. (**E–F**) Extracellular acidification rate (ECAR) analysis of WT or *Chi3l1⁻/⁻* BMDM cells. BMDMs were sequentially treated with glucose, oligomycin, and 2-DG as indicated during seahorse. Unpaired Student's t-test (**D, F**). *p<0.05, **p<0.01, ***p<0.001.

The online version of this article includes the following source data and figure supplement(s) for figure 5:

**Source data 1.** Numerical data of *Figure 5B–F*.

**Figure supplement 1.** Comparison of Kupffer cell (KC) polarization between wild-type (WT) and *Chi3l1⁻/⁻* mice during metabolic dysfunction-associated steatotic liver disease (MASLD) progression.

**Figure supplement 1—source data 1.** Numerical data of *Figure 5—figure supplement 1A*.

**Figure supplement 2.** Recombinant Chi3l1 (rChi3l1) inhibits glucose utilization in *Chi3l1⁻/⁻* bone marrow-derived macrophages (BMDMs).

**Figure supplement 2—source data 1.** Numerical data of *Figure 5—figure supplement 2A–D*.

cell death compared to WT controls in vitro, as indicated by elevated Cl-Casp3 immunostaining (*Figure 6A and B*). This result was further confirmed by increased LDH release from *Chi3l1⁻/⁻* KCs (*Figure 6C*). To delineate the cellular source of Chi3l1 in MASLD livers, we performed immunostaining on serially sectioned liver tissues from mice fed an HFHC for 16 weeks. Consecutive sections were independently probed for Chi3l1 or lineage-specific markers (HNF4α, Desmin, Iba1), enabling cellular localization through morphological alignment across sequential slices. This revealed predominant Chi3l1 expression in hepatic macrophages (*Figure 6—figure supplement 1A*). Moreover, Chi3l1 expression was significantly elevated in KCs (F4/80⁺TIM4⁺) of HFHC-fed mice, compared to NCD controls (*Figure 6—figure supplement 1B*), suggesting Chi3l1 is primarily expressed in KCs and may function as a KC-autonomous regulator.

To determine cell-autonomous effects, we generated KC-specific *Chi3l1* knockout mice (Chi3l1-KpKO mice) and analyzed their response to HFHC feeding. Flow cytometry of NPCs revealed a significantly lower KC population (CD45⁺ F4/80ʰⁱ CD11bˡᵒʷ TIM4⁺) in Chi3l1-KpKO mice compared to *Clec4f cre* controls after 16 weeks of HFHC (*Figure 6D and E*). During this analysis, we observed no reduction in KCs in *Clec4f cre* control mice, raising the possibility that Cre insertion itself might influence KC maintenance. To test this, we performed co-staining for TIM4 and Ki67, which revealed significantly higher numbers of proliferating KCs in *Clec4f cre* mice compared with C57BL/6J mice under NCD. Notably, following HFHC feeding, *Clec4f cre* mice exhibited an even greater increase in KC proliferation (*Figure 6—figure supplement 2A and B*), suggesting that cre insertion enhanced KC self-renewal, which contributes to maintenance of the KC pool in these mice under MASLD. Despite this proliferative baseline, Chi3l1-KpKO mice still displayed a reduced KC number (*Figure 6D and E*) and increased KC death. This was corroborated by TIM4/TUNEL co-staining, which showed elevated KC death in Chi3l1-KpKO livers (*Figure 6F and G*). Collectively, these findings demonstrate that genetic ablation of *Chi3l1* amplifies glycolytic flux and sensitizes KCs to lipotoxic stress in vitro, and drives KC depletion through accelerated cell death in vivo. Although Chi3l1 is not exclusively expressed by KCs, these data establish it as a critical regulator of KC survival in MASLD, likely acting in a cell-autonomous manner within the KC population.

## Discussion

This study identifies KC death as an early and selective pathological feature of MASLD across multiple dietary models. Compared with other hepatic cell populations, KCs exhibit markedly increased susceptibility to apoptosis during early disease. Through metabolomic and functional analyses, we demonstrate that KCs undergo progressive metabolic reprogramming characterized by sustained activation of glycolysis. Importantly, our data support a causal role for excessive glycolytic flux in driving KC death. Pharmacologic glycolytic stimulation (high glucose or PDK1 activation with PS48), enforced glycolytic dependence (oligomycin), and KC-specific Chi3l1 deletion each accelerated KC apoptosis, whereas glycolytic inhibition (2-DG) was protective. Together, these findings establish hyperactivation of glycolysis as a central mechanism underlying KC vulnerability and depletion in MASLD (*Figure 7*).

KCs loss has been reported to varying degrees across MASLD models, reflecting differences in diet composition, disease duration, macrophage heterogeneity, and marker selection. For example, in a 6-week MCD model, approximately 10% of CLEC4F⁺ KCs were TUNEL⁺, indicating apoptotic death

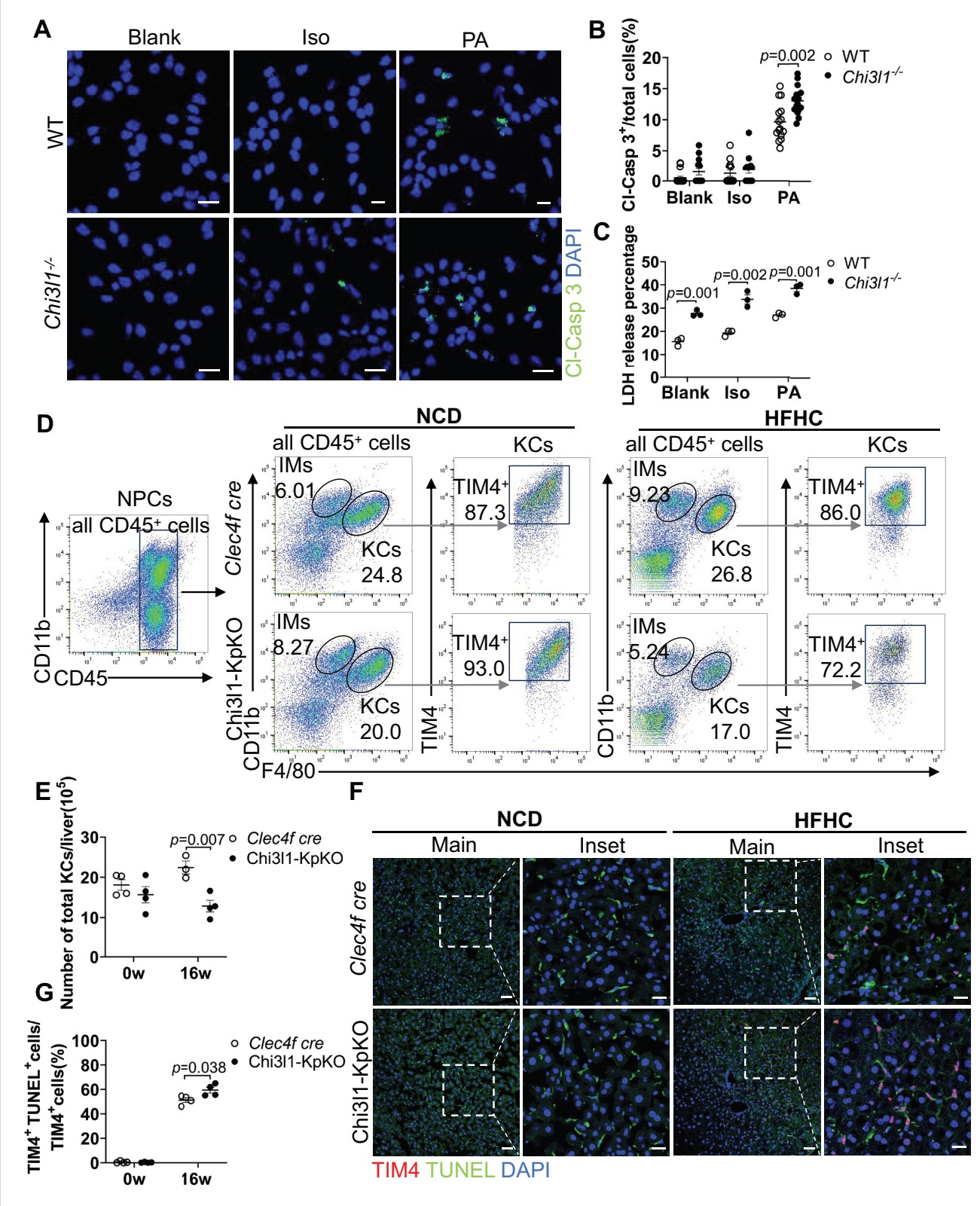

**Figure 6.** Enhanced glycolysis accelerated Kupffer cell (KC) death during metabolic dysfunction-associated steatotic liver disease (MASLD). (**A**) Cleaved caspase 3 (Cl-Casp3) staining to detect wild-type (WT) and *Chi3l1*[-/-] KC death. Cells were under treatment without (blank) or with either isopropyl alcohol (Iso) or palmitic acid (PA) for 24 hr. Scale bar: 20 μm. (**B**) Cl-Casp3+ cells were quantified. (**C**) Lactate dehydrogenase (LDH) release measurement in culture medium of KCs isolated from male WT and *Chi3l1*[-/-] mice was measured after treatment for 24 hr with: blank (no treatment), ISO (vehicle control),

*Figure 6 continued on next page*

*Figure 6 continued*

800 µM PA. (**D**) Flow cytometry analysis of KCs (CD45[+] F4/80[hi] CD11b[low] TIM4[+]) and monocyte-derived macrophages (MoMFs) (CD45[+] F4/80[low] CD11b[hi] TIM4[-]) among nonparenchymal cells (NPCs) in *Clec4f-cre* and Chi3l1-KpKO mice fed high-fat high-cholesterol diet (HFHC) for 0 or 16 weeks. (**E**) KC counts were quantified. n=4 mice/group. (**F**) KC death was assessed by immunostaining of TIM4 (KC marker, green), TdT-mediated dUTP Nick-End Labeling (TUNEL) (red), and DAPI (nuclei, blue) in liver sections from *Clec4f-cre* and Chi3l1-KpKO mice fed HFHC for 0 or 16 weeks. Scale bar: 50 µm (main panels) and 20 µm (inset). (**G**) KC death was quantified. n=4 mice/group. Representative images shown (**A, D, F**). Unpaired Student's t-test (**B, C, E, G**). p-Value as indicated.

The online version of this article includes the following source data and figure supplement(s) for figure 6:

**Source data 1.** Numerical data of *Figure 6B–C, E and G*.

**Figure supplement 1.** Chi3l1 is majorly expressed in hepatic macrophages, and its expression is upregulated during metabolic dysfunction-associated steatotic liver disease (MASLD).

**Figure supplement 1—source data 1.** Numerical data of *Figure 6—figure supplement 1B*.

**Figure supplement 2.** Cre insertion promotes Kupffer cell (KC) self-proliferation.

**Figure supplement 2—source data 1.** Numerical data of *Figure 6—figure supplement 2B*.

(*Tran et al., 2020*). In contrast, another 6-week MCD study reported a marked reduction in TIM4[+] KCs, from 66% to 26% (*Zhang et al., 2024*). More moderate KC loss has been observed in HFD models, with an ~20% reduction of TIM4[+] KCs after 16 weeks (*Daemen et al., 2021*), whereas prolonged Western diet feeding resulted in a >50% decrease in TIM4[+] KCs at 36 weeks (*Daemen et al., 2021*). Together, these studies underscore the model-dependent nature of KC loss and highlight the importance of

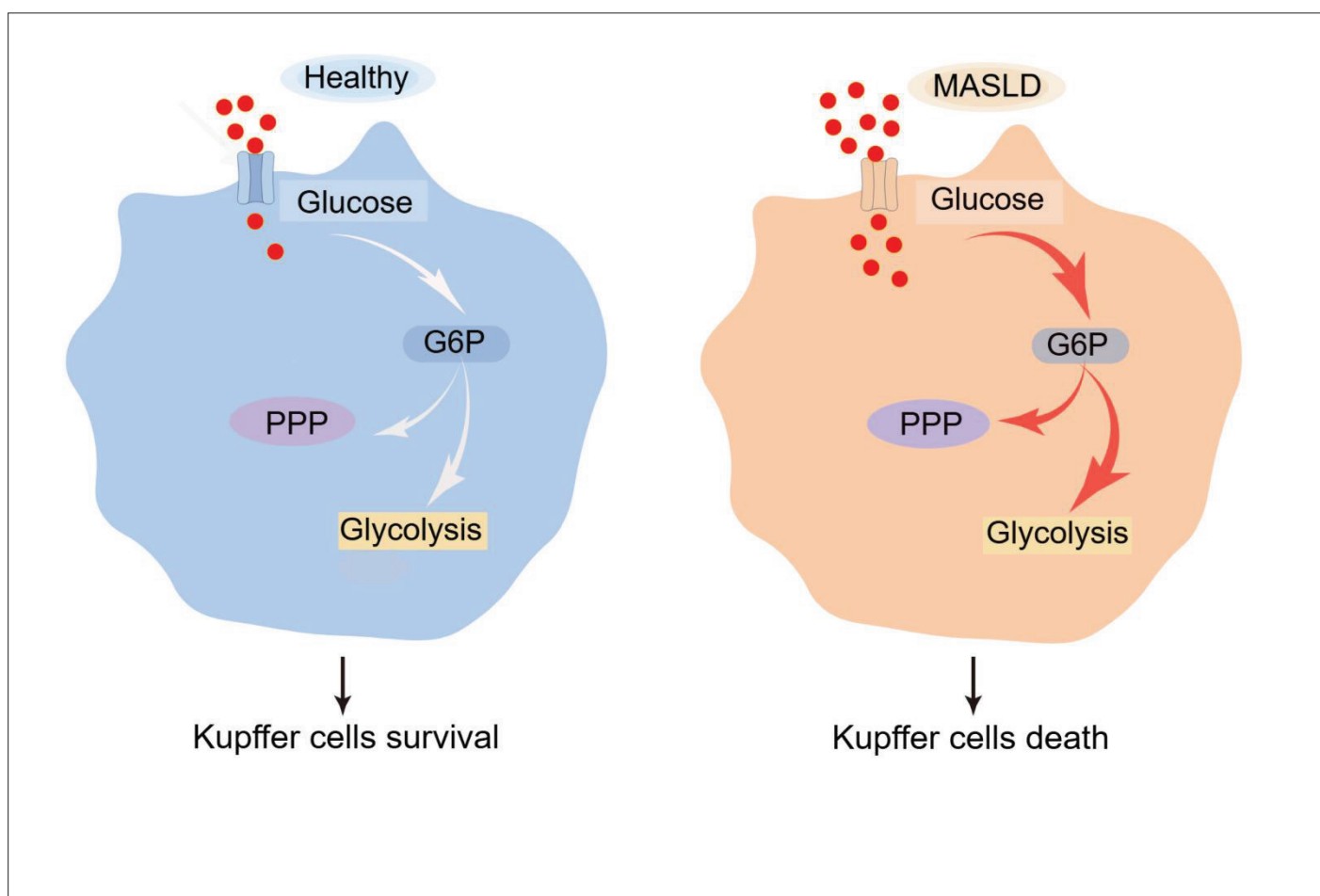

**Figure 7.** Excessive glycolysis enhancement promotes Kupffer cell (KC) death in metabolic dysfunction-associated steatotic liver disease (MASLD). (Left) Under physiological conditions, KCs maintain basal glucose metabolism supporting cellular homeostasis and survival. (Right) During MASLD progression, KCs undergo excessive glycolysis enhancement, which accelerates KC death.

experimental context and marker selection when assessing KC dynamics in MASLD. Importantly, our spatial analyses further reveal preferential periportal KC death during early MASLD. Given the metabolic zonation of the liver and higher periportal glucose flux, this regional susceptibility supports a mechanistic link between glucose metabolism and KC survival. These findings suggest that metabolic microenvironment, in addition to inflammatory signaling, shapes KC fate during MASLD progression.

While glycolytic reprogramming is classically associated with macrophage activation (*Tacke, 2017*), our data reveal a critical distinction in tissue-resident KCs. In MASLD, sustained and excessive glycolytic activation does not simply maintain inflammatory function but instead drives apoptotic loss. Thus, glycolysis in this context becomes maladaptive, serving as a determinant of cell fate rather than merely an activation program. This shifts the conceptual framework from inflammation-centered KC dysfunction to metabolism-driven KC attrition. Our findings also clarify the context-dependent role of Chi3l1 in liver disease. Prior studies in advanced MASH and fibrotic models have described pro-fibrotic functions of Chi3l1, including activation of HSCs and modulation of macrophage survival pathways (*Kim et al., 2023*, *Higashiyama et al., 2019*, *Nishimura et al., 2021*). In contrast, using an HFHC model that recapitulates steatohepatitis without fibrosis, we identify a distinct early-stage role for Chi3l1 in maintaining KC metabolic homeostasis. KC-specific deletion of Chi3l1, but not broader myeloid deletion, accelerated KC loss and worsened steatosis (*He et al., 2025*). These results establish Chi3l1 as a KC-autonomous regulator that restrains glycolytic overload. The apparent duality of Chi3l1 function likely reflects disease stage and cellular source: protective in early MASLD through preservation of resident KCs, but pro-fibrotic in advanced disease when derived from MoMFs and acting on stellate cells.

Our prior work demonstrated that KCs exhibit relative metabolic inflexibility compared with MoMFs, relying heavily on glucose utilization (*He et al., 2025*). The current study extends this concept by showing that loss of Chi3l1 leads to excessive glycolytic flux that exceeds KC metabolic tolerance as supported by $^{13}$C-glucose tracing, enhanced lipotoxic sensitivity in vitro, and accelerated cell death in vivo. Thus, Chi3l1 functions as a metabolic checkpoint that fine-tunes glucose uptake and prevents lethal glycolytic overload. This selective dependence explains why Chi3l1 deficiency disproportionately affects KCs while sparing other macrophage populations.

Several limitations merit consideration. First, although multiple murine dietary models were used, validation in human MASLD—particularly spatial assessment of KC death and metabolic signatures—is necessary. Second, while we identify glycolytic hyperactivation as an upstream driver, the downstream apoptotic mechanisms remain incompletely defined. Potential contributors include oxidative stress, redox imbalance, LA accumulation, or mitochondrial insufficiency under high glycolytic flux. Third, the interplay between glucose metabolism and other metabolic pathways, including fatty acid oxidation, warrants further investigation.

Despite these limitations, our findings have translational implications. Therapeutic strategies that preserve KC metabolic balance—rather than broadly suppressing inflammation—may represent a novel approach for MASLD. Modulation of Chi3l1 signaling or selective targeting of key glycolytic regulators within KCs could help maintain the resident macrophage pool, preserve hepatic immune homeostasis, and potentially delay disease progression. In summary, we identify excessive glycolytic reprogramming as a fundamental mechanism driving selective KC apoptosis in early MASLD, with periportal predominance. Chi3l1 functions as an endogenous metabolic safeguard that restrains this lethal shift. Targeting KC-specific metabolic vulnerability may offer a new strategy to preserve hepatic immune integrity and modify MASLD progression.

## Materials and methods
### Animal experiments and procedures
#### Animals
*Chi3l1*$^{-/-}$ (strain no. T014402), *Chi3l1*$^{flox//flox}$ (strain no. T013652), and *Clec4f-cre* (C-type lectin domain family 4-Cre (*Clec4f-cre*), strain no. T036801) mice with a C57BL/6J background were purchased from GemPharmatech. Accordingly, C57BL/6J mice (strain no. N000013) were used as WT mice. To generate Chi3l1-KpKO mice, *Chi3l1*$^{flox//flox}$ mice were crossed with *Clec4f-cre* mice, and knockout efficiency was examined in KCs previously (*He et al., 2025*). All mouse colonies were maintained at the Animal Core Facility of Yunnan University. The animal studies were approved by the Yunnan University

Institutional Animal Care and Use Committee (IACUC, Approval No. YNU20220314). Male mice aged 6–8 weeks were used in this study. For 2-DG treatment in mice, WT C57BL/6J mice were fed a HFHC for 5 weeks. Beginning at week 3, mice received intraperitoneal injections of either vehicle or 2-DG (50 mg/kg) every other day for 2 weeks. Mice were then euthanized for analysis of KC viability and liver pathology.

## Construction of MASLD/MASH mouse model

Mice were provided an HFHC (Research Diet, d12108c, 40 kcal% fat and 1.25% cholesterol) or an HFD (Research Diet, d12492, 60 kcal% fat). Another group of mice was fed an MCD (Research Diet, A02082002BR). Throughout the feeding period, the body weight and food consumption of the mice were observed and recorded weekly. Once the dietary intervention was completed, the mice were euthanized. Liver and murine serum samples were collected for further analysis. ALT and AST levels in the serum, as well as cholesterol (TC) and TG levels in both serum and liver tissues, were quantified using commercially available kits (Nanjing Jiancheng Bioengineering Institute). Genotyping sample preparation and procedure were conducted as previously described (*He et al., 2025*).

## Isolation of NPCs and KCs

Hepatic NPCs were isolated as previously described (*Chen et al., 2024*). In brief, the abdominal cavity of mice was promptly opened, and the hepatic portal vein was located after anesthesia. A soft needle was inserted for perfusion with perfusion buffer (100 mL 1× HBSS, 200 µL 0.5 mol/L EGTA) for 3–5 min, while the inferior vena cava was cut to clear the liver. Subsequently, digestion buffer (60 mL 1× HBSS, 120 µL 2 M MgSO$_4$, 60 µL 1.25 M CaCl$_2$•H$_2$O, 0.015 g collagenase I) was injected at a consistent rate for approximately 15 min. Following perfusion, the liver was delicately transferred into a pre-cooled Petri dish with PBS, cut into 2–3 mm fragments, and incubated at 37°C for 15 min in digestion buffer. The liver tissue fluid was then filtered through a 70 µm cell filter and centrifuged at 50×$g$ for 5 min (ac/brake = 0) at 4°C to collect the supernatant. This supernatant underwent centrifugation at 450×$g$ for 5 min (ac/brake = 5) at 4°C to obtain the NPC population. The NPCs were suspended in a 15 mL centrifuge tube with 4 mL 20% OptiPrep solution, followed by centrifugation at 3000 rpm for 17 min (ac/brake = 0) at 4°C. After centrifugation, the cells at the boundary between the OptiPrep solution and 1× HBSS buffer were collected into a new tube, supplemented with 1× HBSS buffer, and centrifuged for 5 min at 450×$g$ (ac/brake = 5). Red blood cells were lysed with 1 mL ACK buffer (150 mM NH4Cl, 10 mM KHCO$_3$, 0.1 mM Na$_2$EDTA, pH = 7.2–7.4), neutralized with 1× HBSS buffer, and centrifuged. The cells were resuspended in cell medium and placed in a 3.5 cm Petri dish for 10 min to obtain KCs by washing out un-adherent cells with PBS. The purity and viability of isolated KCs reached 90% and 80%, respectively, which was confirmed by staining with anti-TIM4 antibodies or trypan blue. LDH activity in culture supernatants was measured using a commercial assay kit (Promega, Cat# G1780) according to the manufacturer's instructions. Absorbance was then measured at 490 nm using a microplate reader (Thermo Fisher Scientific, Multiskan Sky).

## KC processing and metabolomic profiling

The extraction method of cellular metabolites was assessed as previously described (*Liu et al., 2022*). Primary KCs were isolated from livers of HFHC-fed mice at 0, 4, and 8 weeks using established protocols. After a 15 min adherence period, cells were washed twice with ice-cold PBS and detached using 1 mL ice-cold PBS followed by gentle scraping. Cell suspensions were transferred to 1.5 mL tubes, and 3×10$^6$ cells were pelleted by centrifugation (1000×$g$, 5 min). Supernatants were discarded, and pellets snap-frozen in liquid nitrogen. Metabolites were extracted from frozen pellets with 80% methanol (vol/vol) and analyzed via LC-MS/MS using an AB Sciex 6500 Plus QTRAP mass spectrometer coupled to an ExionLC system. All metabolomic processing and data analysis were performed by BioDeep (https://www.biodeep.cn). Related data are provided in *Supplementary file 1*.

## Flow cytometry analysis of NPCs

The NPCs were resuspended in fluorescence-activated cell sorting (FACS) buffer, consisting of PBS supplemented with 2% bovine serum albumin. 1 µL of anti-CD16/CD32 antibody (Invitrogen, Cat# 14-0161-86) was added to the cell suspension, and the mixture was incubated at 4°C for 10 min to block any nonspecific binding. After blocking, the mouse NPCs were labeled with monoclonal

antibodies conjugated with fluorescent dyes. This labeling process was carried out at 4°C for 30 min. The labeled cells were washed thrice with cold FACS buffer. The antibodies used for labeling were as follows: anti-mouse F4/80 (APC) (Invitrogen, Cat# 17-4801-82), anti-CD45 (eFluor450) (Invitrogen, Cat# 48-0451-82), anti-mouse TIM4 (PE) (Invitrogen, Cat# 12-5866-82), and anti-mouse CD11b (PerCP/Cyanine5.5) (BioLegend, Cat# 101228). All the primary antibodies were used at a dilution of 1:100. The samples were analyzed by flow cytometry using an LSR Fortessa Cell Analyzer (BD Biosciences). The data obtained were further analyzed using FlowJo version 10.0.

## Preparation of BMDMs

To obtain BMDMs, femur and tibia bone marrow from healthy male C57BL6/J was extracted, resuspended in DMEM/F12 medium (VivaCell, Cat# C3113-0500) containing 10% FBS (VivaCell, Cat# C04001-500) and 20% L929 conditioned medium, seeded in culture dishes, and cultured at 37°C with 5% $CO_2$ in a humidified atmosphere for 7 days. Fresh medium was added on day 4. The cells were maintained in a standard 37°C in 5% $CO_2$ incubator. L929 was a gift from Dr. Guangxun Meng (Hainan Academy of Medical Sciences). Cell identity has been authenticated by the STR profiling. Mycoplasma contamination was tested negative by PCR.

## Isotope tracing

[13]C-tracing experiments were assessed as previously described (*Tong et al., 2024*). For [13]C-tracing experiments, BMDMs were isolated from WT and *Chi3l1*[-/-] mice, cultured to maturity, and replated. After 12 hr, cells were incubated for 12 hr in glucose-depleted medium supplemented with 10% dialyzed FBS and 15 mmol/L universally labeled [U-[13]C]glucose (Cambridge Isotope Laboratories, CLM-1396-1). Cells were subsequently washed twice with ice-cold glucose-free medium. Following supernatant removal, metabolites were extracted using pre-cooled 80% (vol/vol) methanol for cell lysis.

Metabolite separation was performed using a Vanquish UHPLC system (Thermo Fisher Scientific) equipped with an Amide column (Waters). The mobile phase consisted of: (A) 10 mM ammonium acetate and 0.3% ammonia in water, and (B) 10 mM ammonium acetate and 0.3% ammonia in 90% acetonitrile. Metabolites were separated using a linear gradient elution. Metabolites were ionized, and mass spectrometry data were acquired using an Orbitrap Fusion Tribrid mass spectrometer (Thermo Fisher Scientific). Targeted metabolite analysis was performed by LC-MS/MS using a QTRAP 5500+ system (SCIEX) coupled to an ExionLC AD UHPLC system (SCIEX). Isotope tracing analysis and metabolomic data processing were conducted by BioDeep using the BioDeep Platform (https://www.biodeep.cn). Related data are provided in *Supplementary file 2*.

## Histology and immunofluorescence H&E staining

Tissues were fixed with buffered 10% paraformaldehyde (Sangon Biotech, Cat# A500684-0500) overnight at 4°C and embedded in paraffin. Ultrathin tissue slices (5 μm) were prepared and deparaffinized. H&E staining was performed on the tissue sections, and the slides were examined under a microscope (Olympus, BP80).

## Immunofluorescence on frozen section

Immunofluorescence staining was conducted on frozen sections of fresh liver tissues. Initially, the tissues were fixed using 2% paraformaldehyde for 1 hr and subsequently dehydrated overnight in a 30% sucrose solution. The following day, tissue embedding was performed using OCT (SAKURA, Cat# 4583), after which ultrathin sections of 5 μm were sliced. Permeabilization was achieved using 0.02% Triton X-100 for 10 min at room temperature, followed by blocking with 5% normal goat serum (VivaCell, Cat# C2530-0100).

### Co-staining of cleaved caspase 3 with Clec4f

The sections were incubated with primary antibodies, including anti-mouse Clec4f (BioLegend, Cat# 156804, 1:300, Alexa Fluor 647) and Cleaved caspase 3 (Cell Signaling, Cat# 9664S, 1:300). The secondary antibodies used were 488-conjugated Affinipure Goat Anti-Rabbit IgG (H+L) (Jackson ImmunoResearch, 111-545-003, 1:600) to label the Cleaved caspase 3. Nuclei were stained with DAPI

(Beyotime, c1006, prediluted), and images were captured using a confocal laser scanning microscope (ZEISS, LSM900).

## Co-staining of TIM4 and F4/80 with Chi3l1

The sections were incubated with primary antibodies, including anti-mouse TIM4 (BioLegend, Cat# 130008, 1:300, Alexa Fluor 647), anti-mouse F4/80 (BioLegend, Cat# 123140, 1:300, Alexa Fluor 594), and Chi3l1 (Abcam, ab180569, 1:400). The secondary antibodies used were 488-conjugated Affini-pure Goat Anti-Rabbit IgG (H+L) (Jackson ImmunoResearch, 111-545-003, 1:600) to label the Chi3l1. Nuclei were stained with DAPI (Beyotime, c1006, prediluted), and images were captured using a confocal laser scanning microscope (ZEISS, LSM900).

## Co-staining of Clec4f, TIM4, HNF4a, Desmin, IBA1, GS, and Ki67 with TUNEL

TUNEL staining was performed on separate frozen sections of liver tissues. Following fixation with 4% paraformaldehyde and treatment with Proteinase K (20 µg/mL in PBS, BBI, Cat# B600169-0002), the sections were permeabilized with Triton X-100 and blocked with normal goat serum. Subsequently, the sections were incubated with an anti-mouse Clec4f antibody (BioLegend, Cat# 156804, 1:300, Alexa Fluor 647), anti-mouse TIM4 (BioLegend, Cat# 130008, 1:300, Alexa Fluor 647), HNF4α (Abcam, Cat# ab181604, 1:400), Desmin (Proteintech, Cat# 15620-1-AP, 1:400), IBA1 (Fujifilm, Cat# 019-19741, 1:300), GS (Proteintech, Cat# 11037-2-AP, 1:400), and Ki67 (Abcam, Cat# ab15580, 1:300). The secondary antibodies used were 488-conjugated Affinipure Goat Anti-Rabbit IgG (H+L) (Jackson ImmunoResearch, 111-545-003, 1:600), followed by TUNEL staining (Servicebio, Cat# G1502-100T) according to the manufacturer's instructions. Nuclei were counterstained with DAPI, and images were captured using a confocal laser scanning microscope (ZEISS, LSM900).

## Immunohistochemical localization of Chi3l1

Liver frozen sections from male C57BL/6J mice fed an HFHC for 16 weeks were sectioned serially (3 µm). Consecutive sections were independently stained using standard immunofluorescence proto-cols for: Chi3l1 (Abcam, ab180569, 1:400). Lineage markers: HNF4α (Abcam, Cat# ab181604, 1:400), Desmin (Proteintech, Cat# 15620-1-AP, 1:400), Iba1 (Fujifilm, Cat# 019-19741, 1:300). The secondary antibodies used were 488-conjugated Affinipure Goat Anti-Rabbit IgG (H+L) (Jackson ImmunoRe-search, 111-545-003, 1:600). Nuclei were counterstained with DAPI, and images were captured using a confocal laser scanning microscope (ZEISS, LSM900). Cellular localization was assigned by aligning morphological features across sequentially stained sections using nuclear and cytoplasmic landmarks.

## Immunofluorescent staining on KCs

KCs were seeded in 24-well plates and subjected to various treatments. In one set of experiments, cells were treated with either DMSO (1 µL, Sangon Biotech, Cat# A100231-0500) as a control or oligomycin (1 mM in 1 µL, Abcam, Cat# ab141829) for 24 hr. Another experiment involved treatment with DMSO or PS48 (20 mM in 1 µL, Sigma, Cat# P0022) for the same duration. Additionally, BMDMs were treated with different glucose concentrations: 5.5 mM glucose alone, 5.5 mM glucose with PA (800 mM in 1 µL, Sigma, Cat# P0500), or 10 mM glucose with PA, all for 24 hr. The cell slides were washed with cold PBS and fixed with 4% paraformaldehyde in PBS for 10 min at room temperature. The cells were permeabilized with 0.02% Triton X-100 for 10 min at room temperature. After blocking with 5% normal goat serum, cells were incubated with primary antibodies anti-Cleaved caspase 3 (Cell Signaling, Cat# 9664S, 1:300) overnight at 4°C. On the second day, after washing with 0.05% PBST, cells were incubated with 488-conjugated Goat Anti-Rabbit IgG (H+L) (Jackson ImmunoRe-search, 111-545-003, 1:1000) for 1 hr at room temperature. After rinsing with 0.05% PBST, cells were counterstained with DAPI (Beyotime, C1006) and mounted onto slides. Images were captured using a confocal laser scanning microscope (ZEISS, LSM900). The identification of KCs follows the above-mentioned experimental protocol. Primary antibodies against TIM4 (BioLegend, Cat# 130008, 1:300, Alexa Fluor 647) are employed to label KCs.

## Calcein-AM staining on KCs

Live-cell staining using Calcein-AM was conducted following the manufacturer's instructions from a commercially available kit (Beyotime, Cat# C2015M). KCs were seeded in 12-well plates and subjected

to various treatments. In one set of experiments, cells were treated with either DMSO (1 μL, Sangon Biotech, Cat# A100231-0500) as a control or oligomycin (1 mM in 1 μL, Abcam, Cat# ab141829) for 24 hr. Another experiment involved treatment with DMSO or PS48 (20 mM in 1 μL, Sigma, Cat# P0022) for the same duration. Additionally, BMDMs were treated with different glucose concentrations: 5.5 mM glucose alone, 5.5 mM glucose with PA (800 mM in 1 μL, Sigma, Cat# P0500), or 10 mM glucose with PA, all for 24 hr.

### Oil Red O staining
Oil Red O staining was conducted on unfixed frozen sections embedded directly in OCT. Frozen sections of the liver were cut at a thickness of 10 μm. After rinsing with water, the sections were immersed in 60% isopropanol for 2 min. Subsequently, the sections were stained with an Oil Red O staining solution (Solarbio, Cat# IO1720) at 37°C for 10–15 min. Following staining, the sections were immediately placed in 60% isopropanol and washed three to five times to eliminate excess dye solution. Nuclei were counterstained with a hematoxylin staining solution. After rinsing with distilled water, the sections were sealed with glycerol gelatin (Solarbio, Cat# S2150).

### Sirius Red staining
Tissues were fixed with buffered 10% paraformaldehyde (Sangon Biotech, Cat# A500684-0500) overnight at 4°C and embedded in paraffin. Ultrathin tissue slices (5 μm) were prepared and deparaffinized. Sirius Red staining (Solarbio, Cat# G1472) was performed according to the manufacturer's instructions, and the slides were examined under a microscope (Olympus, BP80).

## Spatial analysis of KC death
To evaluate the spatial distribution of KC death along the portal-central axis, the distance between the portal vein (PV) and central vein (CV) was measured in histological sections. The PV-CV axis was systematically divided into three equidistant zones (periportal, intermediate, and pericentral) for regional analysis. KC death was quantified by isolating fluorescence signals corresponding to cell death markers (TUNEL) through channel thresholding and noise reduction. The positive area and integrated fluorescence intensity were measured within each zone, excluding vascular structures to focus on parenchymal KC populations. This zonal approach enabled comparative assessment of KC death patterns across different hepatic microenvironments.

## Diagnosis of MASLD activity score
Murine MASLD activity was assessed histologically using the NAFLD Activity Score (NAS) on H&E-stained liver sections based on the following features: hepatocyte ballooning degeneration (0–2), lobular inflammation (0–3), and steatosis grade (0–3) (*Brunt et al., 2011*). The individual scores were summed to yield the total NAS (range 0–8) per animal. A NAS≥5 was considered diagnostic for steatohepatitis (MASH), NAS≤3 indicated not-MASH, and NAS=4 was indeterminate.

## ECAR measurement
The ECAR measurements were conducted according to the manufacturer's instructions (Agilent, Cat# 103020-100). Briefly, BMDMs were cultured in a Seahorse XF24 cell culture plate. After 12 hr of culture, the DMEM culture medium was pre-treated for 24 hr. 1 hr before the analysis, the culture medium was changed to the corresponding XF basal medium (Agilent, Cat# 103334-100) supplemented with glutamine (the final concentration was 2 mmol/L, Agilent, Cat# 103579-100), and the culture plates were incubated at 37°C without $CO_2$. Compounds were added in the following order: 10 mM glucose, 1.0 mM oligomycin, and 50 mM 2-deoxyglucose. Measurements were conducted using a Seahorse XF24 analyzer (Agilent Technologies). After completing the Seahorse XF Glycolytic Stress Test, basic glycolytic ability and total glycolytic ability were calculated based on the generated report.

## LDH release assay for cytotoxicity measurement
The LDH measurements were conducted according to the manufacturer's instructions (Promega, Cat# G183A). An appropriate amount of cells were inoculated into the culture plate and treated according to the experimental plan. At the scheduled detection time, 50 μL of supernatant from each well was added to a 96-well plate, and then 50 μL of working solution was added to each well, and incubated

at room temperature in the dark for 30 min. Immediately after the reaction, 50 µL of termination solution was added to each well to terminate the reaction. Immediately thereafter, the absorbance of the reaction was measured at 490 nm using an enzyme-labeled instrument (Thermo Scientific).

## Western blot analysis

The sample preparation and procedure were conducted as previously described (*Chen et al., 2024*). The following antibodies were used: anti-Caspase 3 (Cell Signaling Technology, Cat# 9662S, 1:1000), anti-Cleaved caspase 3 (Cell Signaling Technology, Cat# 9664S, 1:1000), anti-β-actin (Proteintech, Cat# 66009-1-Ig, 1:1000). Peroxidase-conjugated Affinipure Goat Anti-Mouse IgG (H+L) (Jackson ImmunoResearch, 115-035-003, 1:2000) and Peroxidase-conjugated Affinipure Goat Anti-Rabbit IgG (H+L) (Jackson ImmunoResearch, 111-035-003, 1:2000) were used for secondary antibody incubation.

## RNA extraction and quantitative real-time PCR

The TRIzol reagent (Invitrogen, Cat# 15596018) was used to lyse cells or ground tissues, and the resulting mixture was centrifuged at 12,000 rpm/min at 4°C for 10 min. The liquid portion (supernatant) was carefully transferred to a new tube. To this tube, 200 µL of chloroform was added and mixed thoroughly. The tube was left at room temperature for 10 min and subsequently centrifuged. The resulting supernatant was collected in a new tube. Isopropanol, in the same volume as the supernatant, was added to the tube and mixed thoroughly. After incubating for 10 min at 4°C, the mixture was centrifuged. The supernatant was then carefully discarded. RNA was precipitated by adding 75% ethanol to the tube, followed by centrifugation. This process was repeated. The tube was dried for 10 min at room temperature, and an appropriate amount of ribonuclease-free water was added to dissolve the RNA precipitate. The concentration and purity of RNA samples were determined using an ultraviolet spectrophotometer (NanoDrop). The RNA template was subjected to reverse transcription using a cDNA first-strand synthesis kit (Takara, Cat# 6210B). Quantitative PCR was performed using SYBR Green Master Mix (Thermo Fisher, Cat# A25742) in triplicates following the manufacturer's instructions. This was performed on a Real-Time PCR QuatStudio1 with accompanying software, following the instructions provided by the manufacturer (Life Technologies, Grand Island, NY, USA). Primer sequences are provided in *Supplementary file 3*.

## Data presentation and statistical analysis

Data in graph figures are presented as mean ± standard error of the mean (SEM). Statistical analyses were performed using SPSS Statistics (version 22). For comparisons between two groups, an unpaired two-tailed Student's t-test was used, while one-way analysis of variance (ANOVA) was applied for comparisons involving three or more groups. A p-value<0.05 was considered statistically significant, with p-values indicated where applicable. All cell culture experiments were repeated at least three times independently. *Figure 7* was created using the Figdraw platform (https://www.figdraw.com).

## Acknowledgements

We thank Dr. Bin Qi (Yunnan University) for suggestions and discussion. We thank Dr. Guangxun Meng (Hainan Academy of Medical Sciences) for providing us with L929 cells. We thank Dr. Cynthia Ju (UTHealth, TX, USA) for advice in manuscript submission. Supported by National Natural Science Foundation of China (32071129 to ZS), Yunnan Provincial Science and Technology Department (C619300A086 to ZS).

## Additional information

### Funding

| Funder | Grant reference number | Author |
|---|---|---|
| National Natural Science Foundation of China | 32071129 to Z.S. | Zhao Shan |

| Funder | Grant reference number | Author |
|---|---|---|
| Yunnan Provincial Science and Technology Department | C619300A086 to Z.S. | Zhao Shan |

The funders had no role in study design, data collection and interpretation, or the decision to submit the work for publication.

## Author contributions

Jia He, Conceptualization, Resources, Data curation, Software, Formal analysis, Validation, Methodology, Writing – original draft, Writing – review and editing; Ran Li, Cheng Xie, Xiane Zhu, Keqin Wang, Data curation; Zhao Shan, Conceptualization, Data curation, Software, Funding acquisition, Methodology, Writing – original draft, Project administration, Writing – review and editing

## Author ORCIDs

Zhao Shan (ID) https://orcid.org/0000-0001-5064-1023

## Ethics

This study was performed in strict accordance with the recommendations in the Guide for the Care and Use of Laboratory Animals of Yunnan University. All animal experiments were approved by the Institutional Animal Care and Use Committee (IACUC) of Yunnan University (Approval No. YNU20220314). All efforts were made to minimize suffering.

Reviewer #3 (Public review): https://doi.org/10.7554/eLife.109206.3.sa1
Reviewer #4 (Public review): https://doi.org/10.7554/eLife.109206.3.sa2
Author response https://doi.org/10.7554/eLife.109206.3.sa3

# Additional files

## Supplementary files

Supplementary file 1. Metabolomics data of primary Kupffer cells from high-fat high-cholesterol diet (HFHC)-fed mice at 0, 4, and 8 weeks.

Supplementary file 2. Isotope tracing and metabolomics data for [U-$^{13}$C]glucose.

Supplementary file 3. qPCR primers.

MDAR checklist

## Data availability

All data generated or analysed during this study are included in the manuscript and supporting files; source data files have been provided.

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

# Appendix 1

## Appendix 1—key resources table

| Reagent type (species) or resource | Designation | Source or reference | Identifiers | Additional information |
|---|---|---|---|---|
| Antibody | Rat monoclonal anti-F4/80 (APC) antibody | Invitrogen | Cat# 17-4801-82; RRID:AB_2784648 | Flow cytometry (1:100) |
| Antibody | Rat monoclonal anti-CD45 (eFluor450) antibody | Invitrogen | Cat# 48-0451-82; RRID:AB_1518806 | Flow cytometry (1:100) |
| Antibody | Rat monoclonal anti-TIM-4 (PE) antibody | Invitrogen | Cat# 12-5866-82; RRID:AB_1257163 | Flow cytometry (1:100) |
| Antibody | Rat monoclonal Anti-CD16/CD32 | Invitrogen | Cat# 14-0161-86; RRID:AB_467135 | Flow cytometry (1:100) |
| Antibody | Rat monoclonal anti-CD11b (PerCP/ Cyanine5.5) antibody | BioLegend | Cat# 101228; RRID:AB_893232 | Flow cytometry (1:100) |
| Antibody | Ly-6G monoclonal antibody (1A8-Ly6g) PE-Cyanine7 | Invitrogen | Cat# 25-9668-82; RRID:AB_2811793 | Flow cytometry (1:100) |
| Antibody | Mouse monoclonal anti-Clec4f (Alexa Fluor 647) antibody | BioLegend | Cat# 156804; RRID:AB_2814082 | IF (1:300) |
| Antibody | Mouse monoclonal anti-TIM4 (Alexa Fluor 647) antibody | BioLegend | Cat# 130008; RRID:AB_2271648 | IF (1:300) |
| Antibody | Mouse monoclonal anti-β-actin antibody | Proteintech | Cat# 66009–1-Ig; RRID:AB_2687938 | WB (1:1000) |
| Antibody | Cleaved caspase 3 rabbit antibody | Cell Signaling Technology | Cat# 9664S; RRID:AB_2070042 | IF (1:300), WB (1:1000) |
| Antibody | Caspase 3 rabbit antibody | Cell Signaling Technology | Cat# 9662S; RRID:AB_331439 | WB (1:1000) |
| Antibody | HNF4α rabbit antibody | Abcam | Cat# ab181604; RRID:AB_2890918 | IF (1:400) |
| Antibody | Desmin Polyclonal antibody | Proteintech | Cat#16520-1-AP | IF (1:400) |
| Antibody | IBA1 rabbit antibody | Fujifilm | Cat# 019-19741; RRID:AB_839504 | IF (1:300) |
| Antibody | Glutamine Synthetase Polyclonal antibody | Proteintech | Cat#11037-2-AP | IF (1:400) |
| Antibody | Ki67 Rabbit antibody | Abcam | Cat# ab15580; RRID:AB_443209 | IF (1:300) |
| Antibody | Rabbit polyclonal Anti-YKL-40/CHI3L1 antibody | Abcam | Cat# ab180569; RRID:AB_2891040 | IF (1:400) |
| Antibody | Peroxidase-conjugated Affinipure Goat Anti-Rabbit IgG (H+L) | Jackson | Cat# 111-035-003; RRID:AB_2313567 | WB (1:2000) |
| Antibody | Peroxidase-conjugated Affinipure Goat Anti-Mouse IgG (H+L) | Jackson | Cat# 115-035-003; RRID:AB_10015289 | WB (1:2000) |
| Antibody | Alexa Fluor 488-conjugated Affinipure Goat Anti-Mouse IgG+IgM (H+L) | Jackson | Cat# 115-545-044; RRID:AB_2338844 | IF (1:600) |
| Antibody | Alexa Fluor 568-goat anti-rabbit IgG (H+L) cross-adsorbed secondary antibody | Invitrogen | Cat# A11011; RRID:AB_143157 | IF (1:600) |
| Chemical compound, drug | FBS | VivaCell | Cat# C04001-500 | |
| Chemical compound, drug | Phosphate Buffered Saline | VivaCell | Cat# C3580-0500 | |
| Chemical compound, drug | DMEM (High glucose) | VivaCell | Cat# C3113-0500 | |

*Appendix 1 Continued on next page*

*Appendix 1 Continued*

| Reagent type (species) or resource | Designation | Source or reference | Identifiers | Additional information |
|---|---|---|---|---|
| Chemical compound, drug | DMEM (No glucose) | Sigma | Cat# D5030 | |
| Chemical compound, drug | $^{13}$C-Glucose | Cambridge Isotope Laboratories | Cat# CLM-1396-1 | |
| Chemical compound, drug | Dialysis fetal bovine serum | Abcam | Cat# DIA0500 | |
| Chemical compound, drug | Sodium pyruvate | Sangon Biotech | Cat# A501259-0100 | |
| Chemical compound, drug | Penicillin-Streptomycin Solution | VivaCell | Cat# C3421-0100 | |
| Chemical compound, drug | Cell Dissociation Solution | Sartorius | Cat# 03-079-1B | |
| Chemical compound, drug | β-Mercaptoethanol | Sigma | Cat# M3148 | |
| Chemical compound, drug | Eosin Y (water soluble) | Aladdin | Cat# E141405 | |
| Chemical compound, drug | Hematoxylin | BBI | Cat# A600701-0050 | |
| Chemical compound, drug | Oil Red O | Solarbio | Cat# IO1720 | |
| Chemical compound, drug | Sirius Red | Sangon Biotech | Cat# A500684-0500 | |
| Chemical compound, drug | High-effect paraffin-ceresin | Shanghai Hualing Rehabilitation Equipment Manufacturing Plant | Cat# N/A | |
| Chemical compound, drug | 10% Neutral Formalin Fix Solution | BBI | Cat# E672001-0500 | |
| Chemical compound, drug | Xylene | Tianjin Zhiyuan Chemical Reagents Co., Ltd. | Cat# N/A | |
| Chemical compound, drug | Neutral balsam | Solarbio | Cat# G8590 | |
| Chemical compound, drug | Isopropanol | Sangon Biotech | Cat# A507048-0500 | |
| Chemical compound, drug | Tissue-Tek OCT compound | SAKURA | Cat# REF:4583 | |
| Chemical compound, drug | Paraformaldehyde | Sangon Biotech | Cat# A500684-0500 | |
| Chemical compound, drug | Acetone | Chron Chemicals | Cat# N/A | |
| Chemical compound, drug | Sucrose | Sangon Biotech | Cat# A502792-0005 | |
| Chemical compound, drug | Triton X-100 | BBI | Cat# A600198-0500 | |
| Chemical compound, drug | Goat serum | VivaCell | Cat# C2530-0100 | |
| Chemical compound, drug | Tween 20 | BBI | Cat# A600560-0500 | |
| Chemical compound, drug | DAPI Staining Solution | Beyotime | Cat# C1006 | |

*Appendix 1 Continued on next page*

*Appendix 1 Continued*

| Reagent type (species) or resource | Designation | Source or reference | Identifiers | Additional information |
|---|---|---|---|---|
| Chemical compound, drug | Antifade Mounting Medium with DAPI | Beyotime | Cat# P0131 | |
| Chemical compound, drug | Omni-Easy One-Step PAGE Gel Fast Preparation Kit | Epizyme | Cat# PG213 | |
| Chemical compound, drug | SDS | BBI | Cat# A600485-0500 | |
| Chemical compound, drug | Glycine | BBI | Cat# A502065-0005 | |
| Chemical compound, drug | Tris | Solarbio | Cat# T8060 | |
| Chemical compound, drug | Methanol | Ghtech | Cat# N/A | |
| Chemical compound, drug | NON-Fat Powdered Milk | BBI | Cat# NON-Fat Powdered Milk | |
| Chemical compound, drug | Collagenase, Type 1 | Diamond | Cat# A004194-0001 | |
| Chemical compound, drug | Cytiva Percoll Centrifugation Media | Cytiva | Cat# 17089101 | |
| Chemical compound, drug | Heparin sodium from Porcine Intestinal | Sangon Biotech | Cat# A603251-0001 | |
| Chemical compound, drug | 1 M HEPES | Solarbio | Cat# H1095 | |
| Chemical compound, drug | OptiPrep | Serumwerk Bernburg | Cat# 1893 | |
| Chemical compound, drug | DNase I, RNase-free | Thermo | Cat# EN0521 | |
| Chemical compound, drug | $CaCl_2$ | GHTECH | Cat#10043-52-4 | |
| Chemical compound, drug | $MgSO_4·7H_2O$ | Sangon Biotech | Cat# A610329-0500 | |
| Chemical compound, drug | TRIzol reagent | Invitrogen | Cat# 15596018 | |
| Chemical compound, drug | UltraPure DNase/RNase-Free Distilled Water | Invitrogen | Cat# 10977015 | |
| Chemical compound, drug | Trichloromethane | Chron Chemicals | Cat# N/A | |
| Chemical compound, drug | PowerUp SYBR Green Master Mix | Applied Biosystems | Cat# A25742 | |
| Chemical compound, drug | DEPC水 | Biosharp | Cat# 701062 | |
| Chemical compound, drug | $MgCl_2$ | GHTECH | Cat# N/A | |
| Chemical compound, drug | KCl | Sangon Biotech | Cat# A501159-0500 | |
| Chemical compound, drug | $NaHCO_3$ | Sangon Biotech | Cat# A500873-0500 | |
| Chemical compound, drug | NaOH | BBI | Cat# A620617-0500 | |

*Appendix 1 Continued on next page*

*Appendix 1 Continued*

| Reagent type (species) or resource | Designation | Source or reference | Identifiers | Additional information |
|---|---|---|---|---|
| Chemical compound, drug | 2-DG | Sigma | Cat# D8375 | |
| Chemical compound, drug | PS48 | Sigma | Cat# P0022 | |
| Chemical compound, drug | Palmitic acid | Sigma | Cat# P0500 | |
| Chemical compound, drug | DMSO | Sangon Biotech | Cat# A100231-0500 | |
| Chemical compound, drug | Proteinase K Solution (20 mg/mL) | BBI | Cat# B600169-0002 | |
| Chemical compound, drug | Glycerol Gelatin aqueous slide mounting medium | Solarbio | Cat# S2150 | |
| Chemical compound, drug | XF basal medium | Agilent | Cat#103334-100 | |
| Chemical compound, drug | XF 200 mmol/L Glutamine solution | Agilent | Cat#103579-100 | |
| Chemical compound, drug | BD Pharmingen Stain Buer (FBS) | BD Biosciences | Cat# 554656 | |
| Chemical compound, drug | Draq7 | BD Biosciences | Cat# 564904 | |
| Chemical compound, drug | High-fat rodent diet with 1.25% cholesterol | Research Diet | Cat# d12108c | |
| Chemical compound, drug | Methionine and choline-deficient diet | Research Diet | Cat# A02082002BR | |
| Chemical compound, drug | Proteinase K Solution (20 mg/mL) | BBI | Cat# B600169-0002 | |
| Chemical compound, drug | Glycerol Gelatin aqueous slide mounting medium | Solarbio | Cat# S2150 | |
| Chemical compound, drug | XF basal medium | Agilent | Cat#103334-100 | |
| Chemical compound, drug | XF 200 mmol/L Glutamine solution | Agilent | Cat#103579-100 | |
| Chemical compound, drug | BD Pharmingen Stain Buer (FBS) | BD Biosciences | Cat# 554656 | |
| Chemical compound, drug | Draq7 | BD Biosciences | Cat# 564904 | |
| Chemical compound, drug | High-fat rodent diet with 1.25%cholesterol | Research Diet | Cat# d12108c | |
| Chemical compound, drug | Methionine and choline-deficient diet | Research Diet | Cat# A02082002BR | |
| Peptide, recombinant protein | Recombinant mouse Chi3l1 | SB | Cat# 50929M08H | |
| Commercial assay or kit | TMR (red) TUNEL Cell Apoptosis Detection Kit | Servicebio | Cat# G1502-100T | |
| Commercial assay or kit | Calcein/PI Cell Viability /Cytotoxicity Assay Kit | Beyotime | Cat# C2015M | |
| Commercial assay or kit | Alanine aminotransferase assay kit | Nanjing Jiancheng Bioengineering Institute | Cat# C009-2-1 | |

*Appendix 1 Continued on next page*

*Appendix 1 Continued*

| Reagent type (species) or resource | Designation | Source or reference | Identifiers | Additional information |
|---|---|---|---|---|
| Commercial assay or kit | Aspartate aminotransferase assay kit | Nanjing Jiancheng Bioengineering Institute | Cat# C010-2-1 | |
| Commercial assay or kit | Total cholesterol assay kit | Nanjing Jiancheng Bioengineering Institute | Cat# A111-1-1 | |
| Commercial assay or kit | Triglyceride assay kit | Nanjing Jiancheng Bioengineering Institute | Cat# A110-1-1 | |
| Commercial assay or kit | Seahorse XF Glycolysis Stress Test Kit | Agilent | Cat# 103020-100 | |
| Commercial assay or kit | PrimeScript II 1st Strand cDNA Synthesis Kit | TaKaRa | Cat# 6210B | |
| Commercial assay or kit | CytoTox 96Non-Radioactive Cytotoxicity Assay | Promega | Cat# G183A | |
| Software, algorithm | GraphPad Prism | GraphPad software | RRID:SCR_002798 | https://www.graphpad.com |
| Software, algorithm | FlowJo V10 | FlowJo Software | RRID:SCR_008520 | https://www.flowjo.com/ |
| Software, algorithm | ImageJ | National Institutes of Health | RRID:SCR_003070 | https://imagej.nih.gov/ij/ |
| Software, algorithm | SPSS | IBM SPSS software | RRID:SCR_002865 | https://www.ibm.com/ |
| Software, algorithm | Seahorse Wave | Agilent Technologies | RRID:SCR_024491 | https://www.agilent.com/ |
| Software, algorithm | ZEN microscope software | ZEISS | RRID:SCR_013672 | https://www.zeiss.com.cn/ |
| Cell line (*Mus musculus*, male) | NCTC clone 929 (L-929) | ATCC | CCL-1 | L929 was a gift from Dr. Guangxun Meng (Hainan Academy of Medical Sciences) |
| Strain, strain background (*Mus musculus*, male) | Chi3l1$^{-/-}$ | GemPharmatech Co., Ltd. | T014402 | Genetic modification: constitutive knockout |
| Strain, strain background (*Mus musculus*, male) | Chi3l1$^{flox/flox}$ | GemPharmatech Co., Ltd. | T013652 | Genetic modification: floxed allele (homozygous) |
| Strain, strain background (*Mus musculus*, male) | Clec4f cre | GemPharmatech Co., Ltd. | T036801 | Genetic modification: Cre recombinase transgene under Clec4f promoter |
| Other | Bone marrow-derived macrophage (BMDM) | This paper | N/A | Strain: C57BL/6J |
| Other | Kupffer cell (KC) | This paper | N/A | Strain: C57BL/6J |

