## [Editor Report · eLife Assessment]

The authors aim to understand why Kupffer cells (KCs) die in metabolic-associated steatotic liver disease (MASLD). This is a **valuable** study using in vitro studies and an in vivo genetic mouse model, suggesting that increased glycolysis contributes to KC death in MASLD. The data presented are now **convincing** and adequately revised. This work will be of interest to researchers in the immunology and metabolism fields.

---

## [Referee Report · Reviewer #3 (Public review)]

This manuscript provides novel insights into altered glucose metabolism and KC status during early MASLD. The authors propose that hyperactivated glycolysis drives a spatially patterned KC depletion that is more pronounced than the loss of hepatocytes or hepatic stellate cells. This concept significantly enhances our understanding of early MASLD progression and KC metabolic phenotype.

Through a combination of TUNEL staining and MS-based metabolomic analyses of KCs from HFHC-fed mice, the authors show increased KC apoptosis alongside dysregulation of glycolysis and the pentose phosphate pathway. Using in vitro culture systems and KC-specific ablation of Chil1, a regulator of glycolytic flux, they further show that elevated glycolysis can promote KC apoptosis.

However, it remains unclear whether the observed metabolic dysregulation directly causes KC death or whether secondary factors, such as low-grade inflammation or macrophage activation, also contribute significantly. Nonetheless, the results, particularly those derived from the Chil1-ablated model, point to a new potential target for the early prevention of KC death during MASLD progression.

The manuscript is clearly written and thoughtfully addresses key limitations in the field, especially the focus on glycolytic intermediates rather than fatty acid oxidation. The authors acknowledge the missing mechanistic link between increased glycolysis and KC death. A few things require clarification.

Strengths:

• The study presents the novel observation of profound metabolic dysregulation in KCs during early MASLD and identifies these cells as undergoing apoptosis. The finding that Chil1 ablation aggravates this phenotype opens new avenues for exploring therapeutic strategies to mitigate or reverse MASLD progression.

• The authors provide a comprehensive metabolic profile of KCs following HFHC diet exposure, including quantification of individual metabolites. They further delineate alterations in glycolysis and the pentose phosphate pathway in Chil1-deficient cells, substantiating enhanced glycolytic flux through 13C-glucose tracing experiments.

• The data underscore the critical importance of maintaining balanced glucose metabolism in both in vitro and in vivo contexts to prevent KC apoptosis, emphasizing the high metabolic specialization of these cells.

• The observed increase in KC death in Chil1-deficient KCs demonstrates their dependence on tightly regulated glycolysis, particularly under pathological conditions such as early MASLD.

Weaknesses:

• The TUNEL staining in the overview in Figure 2 is not convincing. Typically the signal overlaps with DAPI, which is mostly not the case in the figures shown.

• The mechanistic link between elevated glycolytic flux and KC death remains unclear.

• Figure S5: shows deltadelta CT values, not relative values. What are the housekeeping genes? There should be at least 2, and they should not have metabolically related functions such as Gapdh.

• Figure 1C: shows WT and KO gating side by side

• The following point has not been answered: "While BMDMs from Chil1 knockout mice are used to demonstrate enhanced glycolytic flux, it remains unclear whether Chil1 deficiency affects macrophage differentiation itself." Expression of certain genes that indicate function does not show whether BMDMs isolated from these KO mice are fully differentiated. Here, counting BM input/ BMDM output, flow cytometry on BMDMs, morphology etc. should be tested.

---

## [Referee Report · Reviewer #4 (Public review)]

Summary:

In this study, He et al. investigate the mechanisms underlying Kupffer cell (KC) loss during metabolic stress. It has long been observed that embryonically derived KCs decline in obesity and liver disease, a loss that is compensated by monocyte recruitment, although the underlying mechanisms remain unclear. The authors propose that metabolic reprogramming, particularly excessive glycolysis, drives KC death. Using an original murine genetic model to modulate glycolysis, they further demonstrate that enhanced glycolytic activity exacerbates KC damage.

Strengths:

Overall, the study is extremely clearly presented, with a convincing and simple message destined to a vast audience.

Weaknesses:

This manuscript has already undergone one round of revisions in which I was not involved. The authors have tried to address several points raised by the previous reviewers, notably regarding the unexpectedly high level of TUNEL staining observed in KCs. However, I share these concerns expressed by the three reviewers that the reported levels remain difficult to reconcile with the biology. A TUNEL positivity rate of ~60% at week 16 of the HFHC diet would imply massive KC death, which should have led to a near-complete depletion of the KC population, something that is not observed. While I agree that the KC compartment is clearly affected under this dietary challenge, I would strongly encourage the authors to carefully rule out potential technical biases that could account for this implausibly high rate of cell death.

Considering the new in-vivo experiment with 2-DG, it is definitely convincing and clearly adds some value to the full study.

So the full story deserves publication.

---

## [Author Response]

The following is the authors’ response to the original reviews.

**Public Reviews:**

**Reviewer #1 (Public review):**
Summary:The authors aim to investigate the mechanisms underlying Kupffer cell death in metabolic-associated steatotic liver disease (MASLD). The authors propose that KCs undergo massive cell death in MASLD and that glycolysis drives this process. However, there appears to be a discrepancy between the reported high rates of KC death and the apparent maintenance of KC homeostasis and replacement capacity.Strengths:This is an in vivo study.Weaknesses:There are discrepancies between the authors' observations and previous reports, as well as inconsistencies among their own findings.Before presenting the percentage of CLEC4F^+^TUNEL^+^ cells, the authors should have first shown the number of CLEC4F^+^ cells per unit area in Figure 1. At 16 weeks of age, the proportion of TUNEL^+^ KCs is extremely high (~60%), yet the flow cytometry data indicate that nearly all F4/80^+^ KCs are TIMD4^+^, suggesting an embryonic origin. If such extensive KC death occurred, the proportion of embryonically derived TIMD4^+^ KCs would be expected to decrease substantially. Surprisingly, the proportion of TIMD4^+^ KCs is comparable between chow-fed and 16-week HFHC-fed animals. Thus, the immunostaining and flow cytometry data are inconsistent, making it difficult to explain how massive KC death does not lead to their replacement by monocyte-derived cells.

We thank the reviewer for the insightful comment and the opportunity to clarify this important point. To ensure consistency between our methodologies, we replaced Clec4f staining with TIM4 staining results as requested by the reviewer. We first showed the number of TIM4^+^ cells per unit area in Figure 1B. The results showed a significant and progressive loss of TIM4^+^ cells per unit area in the liver parenchyma, decreasing from approximately 60 cells/FOV at baseline (0w) to nearly 50 at 4w and further to about 30 at 16w post-HFHC diet. This finding is fully consistent with our flow cytometry data. The percentage of the embryonically derived KC population (CD11blow F4/80hi TIM4hi) among CD45^+^ cells dropped from 30.2% (0w) to 24.3% (4w) and 17.6% (16w) (Revised Figure 1C). The absolute number per gram of liver decreased from roughly 12 x 10^5^ (1w) to 9 x 10^5^ (4w) and 5 x 10^5^ (16w) (Revised Figure 1D).

These data suggest that despite the reported high rate of cell death among CLEC4F^+^TIMD4^+^ KCs, the population appears to self-maintain, with no evidence of monocyte-derived KC generation in this model, which contradicts several recent studies in the field.

We appreciate the reviewer’s insightful comment. We agree that our data show no substantial generation of monocyte-derived Kupffer cells (MoKCs) within the 16-week HFHC model. However, we do not believe the remaining embryonic KCs(EmKCs) are maintained through self-renewal, as the proportion of Ki67^+^TIM4^+^ cells remains low at all time points (Revised Figure S2D). Instead, our observations align with a phased replacement model: recruited monocytes first differentiate into monocyte-derived macrophages (MoMFs), which we see accumulate (Revised Figure S2B, S2C), and only later adopt a KC phenotype. Consistent with this, our 16-week model shows significant EmKC loss and MoMF expansion, but not yet the emergence of TIM4-MoKCs. This timing is supported by prior studies, where TIM4-KCs were observed at 24 weeks, but not at 16 weeks, on similar diets (Ref. 1,2). Therefore, we interpret our findings as capturing an earlier phase of MASLD progression, characterized by EmKC death and MoMF accumulation, prior to their full differentiation into MoKCs.

Moreover, there is no evidence that TIM4^+^CLEC4F^+^ KCs increase their proliferation rate to compensate for such extensive cell death. If approximately 60% of KCs are dying and no monocyte-derived KCs are recruited, one would expect a much greater decrease in total KC numbers than what is reported.

Thank you for raising this point, which allows for an important clarification. The interpretation that approximately 60% of KCs are dying is correct, but this refers to the proportion of the remaining KC population at 16 weeks that is TUNEL^+^, not to 60% of the original KC pool. Since our data show that over half of the EmKCs are lost by 16 weeks (Revised Figure 1B), the 60% of dying cells at this late time point corresponds roughly to only 25-30% of the total original KC population at baseline. This distinction reconciles the high rate of apoptosis observed late in disease with the overall progressive depletion of the EmKC pool.

It is also unexpected that the maximal rate of KC death occurs at early time points (8 weeks), when the mice have not yet gained substantial weight (Figure 1B). Previous studies have shown that longer feeding periods are typically required to observe the loss of embryo-derived KCs.

We appreciate the reviewer’s insightful observation. We think KC death is a continuous event during MASLD. To induce MASH, previous studies typically assess the loss of EmKCs after longer feeding periods, which might leave us an impression that longer feeding periods are required to observe substantive loss of embryonically derived KCs. In our HFHC model, the proportion of dying KCs was already elevated by 8 weeks, and this high rate was sustained through the 16-week endpoint. In a separate MCD dietary model characterized by rapid MASLD progression, a high rate of KC death was detectable as early as 6 weeks (Revised Figure 1F). Collectively, these data suggest that the onset of significant KC death is dependent on the pace of MASLD pathogenesis, more likely an early-initiated event that is through MASLD progression.

Furthermore, it is surprising that the HFD induces as much KC death as the HFHC and MCD diets. Earlier studies suggested that HFD alone is far less effective than MASH-inducing diets at promoting the replacement of embryonic KCs by monocyte-derived macrophages.

We appreciate the reviewer’s insightful comment. In our study, we observed significant KCs death under both HFD and HFHC feeding for 20, 16 weeks, respectively. Moreover, both HFHC and HFD induced similar stages of MASLD (characterized by significant lipid accumulation without fibrosis development) by these time points (Authir response image 1). Therefore, these data support that the onset of substantial KCs death may be an early MASLD event, before the progression to MASH. Additionally, this finding aligns with existing literature showing that 16 weeks of HFD feeding alone is sufficient to cause a marked reduction in the TIM4^+^KCs population (Ref. 1).

**Author response image 1. sa3fig1:** Detection of liver fibrosis in MASLD mouse models. Male wild-type C57BL/6J mice were fed a high-fat, high-cholesterol diet (HFHC) for 16 weeks or a high-fat diet (HFD) for 20 weeks to induce MASLD. Mice fed a normal chow diet (NCD) served as controls. (A) Sirius Red staining of liver sections was performed to assess collagen deposition and fibrosis during MASLD progression. Scale bar, 20 μm. (B) Western blot analysis of liver tissue lysates showing α-smooth muscle actin (α-SMA) expression as a marker of hepatic stellate cell activation and liver fibrosis.

In Figure 2D, TIMD4 staining appears extremely faint, making the results difficult to interpret. In contrast, the TUNEL signal is strikingly intense and encompasses a large proportion of liver cells (approximately 60% of KCs, 15% of hepatocytes, 20% of hepatic stellate cells, 30% of non-KC macrophages, and a proportion of endothelial cells is also likely affected). This pattern closely resembles that typically observed in mouse models of acute liver failure. Given this apparent extent of cell death, it is unexpected that ALT and AST levels remain low in MASH mice, which is highly unusual.

Thank you for this important feedback. To address concerns about the clarity of our imaging, we have provided high-resolution split-channel raw images for Figure 2D (Revised Figure 2D), which distinctly show the localization of TIM4, TUNEL, and GS. These confirm the progressive reduction of TIM4^+^KCs and the increase in TUNEL^+^ TIM4^+^ cells over time. We agree that the high proportion of TUNEL^+^ cells seems at odds with the modest ALT/AST elevation. This discrepancy might be explained by the distinct nature of cell death in MASLD. Unlike the acute necrosis with membrane rupture seen in acute liver failure—which causes massive, rapid enzyme release— obesity-related liver injury is a chronic process dominated by apoptosis (Ref. 4,5). Apoptosis preserves membrane integrity until late stages (Ref. 6), with dying cells packaged into apoptotic bodies for efficient phagocytic clearance by neighboring macrophages (Ref. 7,8). This controlled disposal system minimizes the leakage of intracellular enzymes. Therefore, the coexistence of widespread apoptosis (high TUNEL signal) with limited enzyme release (low ALT/AST) is a recognized feature of chronic MASLD pathogenesis.

No statistical analysis is provided for Figure 5D, and it is unclear which metabolites show statistically significant changes in Figure 5C.

We thank the reviewer for raising this statistical problem. We have now included statistical analysis in Revised Figure 5D.

In addition, there is no evaluation of liver pathology in Clec4f-Cre × Chil1flox/flox mice. It remains possible that the observed effects on KC death result from aggravated liver injury in these animals. There is also no evidence that Chil1 deficiency affects glucose metabolism in KCs in vivo.

We thank the reviewer for these important points. We previously characterized the liver pathology of Clec4f^ΔChil1^ mice in detail (preprint: eLife 2025, DOI: 10.7554/eLife.107023.1, Fig. 2). On a normal chow diet, these mice showed no differences in body weight, hepatic lipid deposition, metabolic parameters, or glucose tolerance compared to controls. However, on an HFHC diet, Clec4f^ΔChil1^ mice developed significantly worse metabolic and histological phenotypes. Crucially, our in vitro data demonstrate that recombinant Chi3l1 directly reduces KC death (preprint, Fig. 6E-F), indicating that the aggravated MASLD in knockout mice is a consequence of increased KC loss, not its cause.

Regarding glucose metabolism, we have previously shown that Chi3l1 deficiency leads to increased glucose uptake by KCs *in vivo* using the fluorescent glucose analog 2-NBDG. This effect was reversed by supplementing knockout mice with recombinant Chi3l1 (preprint Fig. 6G-H). This provides direct evidence that Chi3l1 modulates glucose uptake in KCs *in vivo*.

Finally, the authors should include a more direct experimental approach to modulate glycolysis in KCs and assess its causal role in KC death in MASH.

We thank the reviewer for this constructive suggestion. To more directly evaluate the role of glycolysis in KCs death *in vivo*, we performed pharmacological inhibition of glycolysis using 2-deoxy-D-glucose (2-DG) in the HFHC-induced MASLD model (Revised Figure 4E–G). Wild-type mice were fed an HFHC diet for four weeks, and 2-DG (50 mg/kg) or vehicle was administered intraperitoneally every other day beginning at week 3. This short intervention period and modest dosing were chosen to limit potential systemic metabolic effects while modulating glycolytic activity during active disease development. KCs apoptosis was assessed by TIM4/TUNEL co-staining. 2-DG treatment significantly reduced the proportion of TUNEL^+^KCs compared with vehicle controls, indicating protection against KCs death. These data together with our complementary in vitro gain-of-function experiments, support a contributory role for excessive glycolytic activity in promoting KC apoptosis in MASLD. We have incorporated these findings into the revised manuscript to strengthen the causal link between glycolytic reprogramming and KCs loss *in vivo* (Revised manuscript, page 7, line 267-282).

**Reviewer #2 (Public review):**
Summary:In this manuscript, He et al. set out to investigate the mechanisms behind Kupffer Cell death in MASLD. As has been previously shown, they demonstrate a loss of resident KCs in MASLD in different mouse models. They then go on to show that this correlates with alterations in genes/metabolites associated with glucose metabolism in KCs. To investigate the role of glucose metabolism further, they subject isolated KCs in vitro to different metabolic treatments and assess cleaved caspase 3 staining, demonstrating that KCs show increased Cl. Casp 3 staining upon stimulation of glycolysis. Finally, they use a genetic mouse model (Chil1KO) where they have previously reported that loss of this gene leads to increased glycolysis and validate this finding in BMDMs (KO). They then remove this gene specifically from KCs (Clec4fCre) and show that this leads to increased macrophage death compared with controls.Strengths:As we do not yet understand why KCs die in MASLD, this manuscript provides some explanation for this finding. The metabolomics is novel and provides insight into KC biology. It could also lead to further investigation; here, it will be important that the full dataset is made available.Weaknesses:Different diets are known to induce different amounts of KC loss, yet here, all models examined appear to result in 60% KC death. One small field of view of liver tissue is shown as representative to make these claims, but this is not sufficient, as anything can be claimed based on one field of view. Rather, a full tissue slice should be included to allow readers to really assess the level of death.

Thank you for raising this point regarding data presentation. We analyzed full tissue slices and found that including a view of the entire slice at a standard magnification makes individual KC difficult to resolve (Author response image 2). To clearly represent the extent and distribution of KCs death across the liver tissue slice, we now include lower-magnification images that provide a wider field of view, allowing readers to assess the pattern across a larger tissue area (Revised Figures 1, 2, 6F).

**Author response image 2. sa3fig2:** Assessment of KCs death on full liver tissue slice. (A) Immunofluorescence staining was performed to detect Kupffer cell (KC) death in liver sections from mice fed an MCD diet for 6 weeks. Cell death was assessed by TUNEL staining (green), and KCs were identified by TIM4 staining (red). Nuclei were counterstained with DAPI (blue). Representative whole-tissue view is shown. Scale bars, 1mm.

Additionally, there is no consistency between the markers used to define KCs and moMFs, with CLEC4F being used in microscopy, TIM4 in flow, while the authors themselves acknowledge that moKCs are CLEC4F+TIM4-. As moKCs are induced in MASLD, this limits interpretation. Additionally, Iba1 is referred to as a moMF marker but is also expressed by KCs, which again prevents an accurate interpretation of the data. Indeed, the authors show 60% of KCs are dying but only 30% of IBA1+ moMFs, as KCs are also IBA1+, this would mean that KCs die much more than moMFs, which would then limit the relevance of the BMDM studies performed if the phenotype is KC specific. Therefore, this needs to be clarified.

We thank the reviewer for the constructive comments. For consistency, we have standardized our KC marker to TIM4 for all immunostaining data, aligning it with our flow cytometry analysis (Revised Figures 1, 2D, 6F). We have also clarified that IBA1 is expressed by hepatic macrophages (both KCs and MoMFs)(Revised Figure 2C, Revised manuscript, page 5, lines 182-183). Moreover, we also included the clarification that 60% of TIM4^+^ KCs are TUNEL^+^ versus 30% of total IBA1^+^ cells further supports that KCs undergo death more readily than MoMFs (Revised manuscript, page 5, lines 186-189). We also acknowleged the limitation of BMDM studies in the Revised manuscript, page 8, line 332-340.

The claim that periportal KCs die preferentially is not supported, given that the majority of KCs are peri-portal. Rather, these results would need to be normalised to KC numbers in PP vs PC regions to make meaningful conclusions.

We thank the reviewer for this important point. We included the normalized data. At 8 weeks, the normalized death rate was significantly higher in periportal versus pericentral regions (p = 0.041), supporting increased periportal KC susceptibility during early MASLD. By 16 weeks, proportional death rates became comparable between zones (Revised Figure 2D, Revised manuscript, page 6, lines 194-201).

Additionally, KCs are known to be notoriously difficult to keep alive in vitro, and for these studies, the authors only examine cl. Casp 3 staining. To fully understand that data, a full analysis of the viability of the cells and whether they retain the KC phenotype in all conditions is required.

We appreciate the reviewer’s suggestions. To confirm the identity and health of isolated KCs in our in vitro studies, we showed that ~95% of primary isolated KCs are TIM4^+^ (Revised Figure S3A). Furthermore, Calcein-AM staining confirmed that the remaining KCs under our experimental conditions are viable and healthy (Revised Figure S4A).

Finally, in the Cre-driven KO model, there does not seem to be any death of KCs in the controls (rather numbers trend towards an increase with time on diet, Figure 6E), contrary to what had been claimed in the rest of the paper, again making it difficult to interpret the overall results.

We thank the reviewer for this comment. During our analysis, we indeed observed no reduction in KCs in the Clec4f cre control mice. This prompted us to consider that Cre insertion itself might influence KCs mainteinence. To investigate this, we performed TIM4/Ki67 co-staining, which revealed significantly higher numbers of proliferating KCs in Clec4f cre mice compared with C57BL/6J mice under NCD. Following HFHC feeding, KCs proliferation in Clec4f cre mice increased even further. These results indicate that Cre insertion enhanced KCs self-renewal in Clec4f cre mice，which contributes to maintenance of the KCs pool during MASLD (Revised Figures S8A and S8B). (Revised manuscript, page 9, line 363-370).

Additionally, there is no validation that the increased death observed in vivo in KCs is due to further promotion of glycolysis.

We thank the reviewer for this constructive suggestion. To more directly evaluate the role of glycolysis in KCs death *in vivo*, we performed pharmacological inhibition of glycolysis using 2-deoxy-D-glucose (2-DG) (Revised Figure 4E–G). Wild-type mice were fed an HFHC diet for five weeks, and 2-DG (50 mg/kg) or vehicle was administered intraperitoneally every other day beginning at week 3. This short intervention period and modest dosing were chosen to limit potential systemic metabolic effects while modulating glycolytic activity in KCs. KCs apoptosis was assessed by TIM4/TUNEL co-staining. 2-DG treatment significantly reduced the proportion of TUNEL^+^KCs compared with vehicle controls, indicating protection against KCs death. These data, together with our complementary in vitro gain-of-function experiments support a contributory role for excessive glycolytic activity in promoting KCs death in MASLD. We have incorporated these findings into the revised manuscript to strengthen the causal link between glycolytic reprogramming and KCs loss *in vivo* (Revised manuscript, page 7, line 267-282).

**Reviewer #3 (Public review):**
This manuscript provides novel insights into altered glucose metabolism and KC status during early MASLD. The authors propose that hyperactivated glycolysis drives a spatially patterned KC depletion that is more pronounced than the loss of hepatocytes or hepatic stellate cells. This concept significantly enhances our understanding of early MASLD progression and KC metabolic phenotype.Through a combination of TUNEL staining and MS-based metabolomic analyses of KCs from HFHC-fed mice, the authors show increased KC apoptosis alongside dysregulation of glycolysis and the pentose phosphate pathway. Using in vitro culture systems and KC-specific ablation of Chil1, a regulator of glycolytic flux, they further show that elevated glycolysis can promote KC apoptosis.However, it remains unclear whether the observed metabolic dysregulation directly causes KC death or whether secondary factors, such as low-grade inflammation or macrophage activation, also contribute significantly. Nonetheless, the results, particularly those derived from the Chil1-ablated model, point to a new potential target for the early prevention of KC death during MASLD progression.The manuscript is clearly written and thoughtfully addresses key limitations in the field, especially the focus on glycolytic intermediates rather than fatty acid oxidation. The authors acknowledge the missing mechanistic link between increased glycolysis and KC death. Still, several interpretations require moderation to avoid overstatement, and certain experimental details, particularly those concerning flow cytometry and population gating, need further clarification.Strengths:(1) The study presents the novel observation of profound metabolic dysregulation in KCs during early MASLD and identifies these cells as undergoing apoptosis. The finding that Chil1 ablation aggravates this phenotype opens new avenues for exploring therapeutic strategies to mitigate or reverse MASLD progression.(2) The authors provide a comprehensive metabolic profile of KCs following HFHC diet exposure, including quantification of individual metabolites. They further delineate alterations in glycolysis and the pentose phosphate pathway in Chil1-deficient cells, substantiating enhanced glycolytic flux through 13C-glucose tracing experiments.(3) The data underscore the critical importance of maintaining balanced glucose metabolism in both in vitro and in vivo contexts to prevent KC apoptosis, emphasizing the high metabolic specialization of these cells.(4) The observed increase in KC death in Chil1-deficient KCs demonstrates their dependence on tightly regulated glycolysis, particularly under pathological conditions such as early MASLD.Weaknesses:(1) The novelty is questionable. The presented work has considerable overlap with a study by the same lab, which is currently under review (citation 17), and it should be considered whether the data should not be presented in one paper.

We appreciate the reviewer for the opportunity to clarify the relationship between the two studies. In our previous work (citation 17), we focused on the transcriptional metabolic differences between Kupffer cells (KCs) and monocyte-derived macrophages (MoMFs) and identified Chi3l1 as a selective protective factor that limits glucose uptake and shields KCs from metabolic stress–induced cell death, with minimal effects on MoMFs. That study directly motivated the current work. The observation that KCs are uniquely protected from metabolic stress led us to hypothesize that excessive glycolytic activation itself may be a primary driver of KCs death, which forms the central question of the present study. Accordingly, the current manuscript shifts the focus from Chi3l1-mediated protection to the mechanistic role of hyperglycolysis in driving KCs mortality, using distinct experimental approaches and addressing a different biological question. Because the two studies address conceptually distinct aims—one defining a protective regulator of KCs survival and the other dissecting glycolysis-driven KCs death mechanisms—we believe they are best presented as separate manuscripts. Combining them into a single study would dilute the mechanistic depth and clarity of each story.

(2) The authors report that 60% of KCs are TUNEL-positive after 16 weeks of HFHC diet and confirm this by cleaved caspase-3 staining. Given that such marker positivity typically indicates imminent cell death within hours, it is unexpected that more extensive KC depletion or monocyte infiltration is not observed. Since Timd4 expression on monocyte-derived macrophages takes roughly one month to establish, the authors should consider whether these TUNEL-positive KCs persist in a pre-apoptotic state longer than anticipated. Alternatively, fate-mapping experiments could clarify the dynamics of KC death and replacement.

We thank the reviewer for this astute observation. As shown in revised Figure 2D, the proportion of TIM4^+^TUNEL^+^KCs peaks at 8 weeks after HFHC feeding and remains elevated at 16 weeks. However, examination of the corresponding single-channel TIM4 staining during this period reveals that the overall density of TIM4^+^ KCs does not undergo abrupt or synchronous depletion. This temporal dissociation between sustained TUNEL positivity and relatively gradual KCs loss suggests that TUNEL-positive KCs do not undergo immediate clearance. Based on these observations, we agree with the reviewer that a substantial fraction of TUNEL-positive KCs likely persists in a prolonged pre-apoptotic or stressed state rather than undergoing rapid cell death. This interpretation is consistent with the absence of extensive KCs depletion or compensatory monocyte infiltration at these time points. Importantly, previous studies (Ref. 1,2) indicate that KCs are eventually lost as MASLD progresses, supporting the notion that KC death is a gradual process that unfolds over an extended time frame rather than acutely.

(3) The mechanistic link between elevated glycolytic flux and KC death remains unclear.

We thank the reviewer for this constructive suggestion. To more directly evaluate the role of glycolysis in KCs death *in vivo*, we performed pharmacological inhibition of glycolysis using 2-deoxy-D-glucose (2-DG) (Revised Figure 4E–G). Wild-type mice were fed an HFHC diet for five weeks, and 2-DG (50 mg/kg) or vehicle was administered intraperitoneally every other day beginning at week 3. This short intervention period and modest dosing were chosen to limit potential systemic metabolic effects while modulating glycolytic activity of KCs. KCs apoptosis was assessed by TIM4/TUNEL co-staining. 2-DG treatment significantly reduced the proportion of TUNEL^+^KCs compared with vehicle controls, indicating protection against KCs death. These data, together with our complementary in vitro gain-of-function experiments, support a contributory role for excessive glycolytic activity in promoting KC apoptosis in MASLD. We have incorporated these findings into the revised manuscript to strengthen the causal link between glycolytic reprogramming and KCs loss *in vivo* (Revised manuscript, page 7, line 267-282).

(4) The study does not address the polarization or ontogeny of KCs during early MASLD. Given that pro-inflammatory macrophages preferentially utilize glycolysis, such data could provide valuable insight into the reason for increased KC death beyond the presented hyperreliance on glycolysis.

We thank the reviewer for this insightful comment. Regarding KCS ontogeny, flow cytometry analysis (Revised Figure 1C) shows that KCs remain uniformly TIM4^hi^ during early MASLD, indicating that monocyte-derived KCs (TIM4^low^) have not yet emerged at these stages. To address KCs polarization, we assessed the expression of M1-type (pro-inflammatory) markers (Nos2, Cxcl9, CIITA, Cd86, Ccl3, and Ccl5) and M2-type (anti-inflammatory) markers (Chil3, Retnla, Arg1, and Mrc1) in KCs isolated from WT mice fed a HFHC diet for 0, 8, and 16 weeks. As shown in revised Figure S5A, M1 markers progressively increase over time, whereas M2 markers remain unchanged or slightly decrease. This polarization shift is consistent with the increased glycolytic activity observed in KCs during early MASLD. Together, these data indicate that embryonically derived KCs undergo a pro-inflammatory polarization accompanied by enhanced glycolytic metabolism during early MASLD, providing mechanistic context for their increased susceptibility to metabolic stress–induced cell death beyond hyperreliance on glycolysis alone (Revised manuscript, page 7-8, line 307-321).

(5) The gating strategy for monocyte-derived macrophages (moMFs) appears suboptimal and may include monocytes. A more rigorous characterization of myeloid populations by including additional markers would strengthen the study's conclusions.

We thank the reviewer for raising this important point. To improve the rigor of our analysis, we adopted gating strategies established in previous studies (PMID: 41131393; PMID: 32562600). Specifically, Kupffer cells were defined as CD45^+^CD11b^+^F4/80^hi^ TIM4^hi^ cells, while monocyte-derived macrophages (MoMFs) were defined as CD45^+^Ly6G^-^CD11b^+^F4/80^low^ TIM4^low/−^ cells, thereby excluding contaminating neutrophils and minimizing inclusion of circulating monocytes. Using this refined gating strategy, we observed a progressive reduction of KCs accompanied by a corresponding increase in MoMFs in WT mice during HFHC feeding (Revised Figures 1C and S2B–C), (Revised manuscript, page 4, line 154-163).

(6) While BMDMs from Chil1 knockout mice are used to demonstrate enhanced glycolytic flux, it remains unclear whether Chil1 deficiency affects macrophage differentiation itself.

We thank the reviewer for this important question. To determine whether Chi3l1 deficiency affects macrophage differentiation, we analyzed the expression of M1-type (pro-inflammatory) markers (Nos2, Cxcl9, CIITA, Cd86, Ccl3, and Ccl5) and M2-type (anti-inflammatory) markers (Chil3, Retnla, Arg1, and Mrc1) in Kupffer cells isolated from WT and *Chil1^-/-^* mice fed a HFHC diet for 0, 8, and 16 weeks. At baseline (0 weeks), Chi3l1 deficiency was associated with elevated expression of multiple M1 markers, whereas M2 marker expression was comparable between WT and *Chil1^-/-^* KCs. During MASLD progression, the pro-inflammatory signature in *Chil1^-/-^* KCs was further enhanced, while anti-inflammatory marker expression became dysregulated (revised Figure S5C). Together, these data indicate that Chi3l1 deficiency does not impair macrophage differentiation per se but biases KCs toward a partially pro-inflammatory, M1-like phenotype, providing additional context for the enhanced glycolytic flux observed in Chi3l1-deficient macrophages (Revised manuscript, page 7-8, line 307-321).

(7) The authors use the PDK activator PS48 and the ATP synthase inhibitor oligomycin to argue that increased glycolytic flux at the expense of OXPHOS promotes KC death. However, given the high energy demands of KCs and the fact that OXPHOS yields 15-16 times more ATP per glucose molecule than glycolysis, the increased apoptosis observed in Figure 4C-F could primarily reflect energy deprivation rather than a glycolysis-specific mechanism.

We thank the reviewer for highlighting this important point. We agree that KCs are highly metabolically active and that perturbations of OXPHOS can influence overall cellular energy balance. As noted in our response to comment #3, we further performed glycolysis inhibition assay by 2-DG *in vivo*, the protection of KCs observed following 2-DG *in vivo* (Revised Figure 4E-G) further provides evidence that increased glycolytic flux is not merely correlated with, but functionally contributes to KCs loss in

MASLD.

(8) In Figure 1C, KC numbers are significantly reduced after 4 and 16 weeks of HFHC diet in WT male mice, yet no comparable reduction is seen in Clec4Cre control mice, which should theoretically exhibit similar behavior under identical conditions.

We thank the reviewer for this comment. During our analysis, we indeed observed no reduction in KCs in the Clec4f cre control mice. This prompted us to consider that Cre insertion itself might influence KCs mainteinence. To investigate this, we performed TIM4/Ki67 co-staining, which revealed significantly higher numbers of proliferating KCs in Clec4f cre mice compared with C57BL/6J mice under NCD. Following HFHC feeding, KCs proliferation in Clec4f cre mice increased even further. These results indicate that Cre insertion enhanced KCs self-renewal in Clec4f cre mice，which contributes to maintenance of the KCs pool during MASLD (Revised Figures S8A and S8B). (Revised manuscript, page 9, line 363-370).

**Recommendations for the authors:**

**Reviewer #2 (Recommendations for the authors):**
To address the concerns raised in the public review, the authors should:(1) Reassess their conclusions using the same panels in flow and microscopy, e.g., the combination of CLEC4F, TIM4, and IBA1. This will allow resKCs (CLEC4F+TIM4+IBA1+), moKCs (CLEC4F+TIM4-IBA1+), and moMFs (CLEC4F-TIM4-IBA1+) to be accurately defined and hence their viability and numbers correctly assessed.

We thank the reviewer for this insightful suggestion. In our flow cytometry analysis, we did not detect a CD45^+^CD11b^low^F4/80^hi^TIM4^low^ population, indicating that monocyte-derived KCs (moKCs) have not emerged in our model at this stage. To more accurately quantify resident KCs (resKCs) in the current study, we replaced CLEC4F with TIM4 staining and enumerated TIM4^+^as well as TIM4^+^TUNEL^+^ cells. These data were highly consistent with CLEC4F^+^TUNEL^+^ cell counts, confirming that moKCs are not involved in KCs death during early MASLD (Revised Figure 1A,B,E,F).

(2) Investigate why the number of KCs in controls and MASLD are so distinct between Figures 1 and 6.

We appreciate the reviewer’s suggestions. Like we explained above, Cre insertion promotes KCs self-renewal (Revised manuscript, Figure S8). This enhanced proliferative capacity likely accounts for the relative preservation of KCs numbers in Clec4f-Cre mice during HFHC feeding, explaining the apparent discrepancy with WT mice (Revised manuscript, Figure 6D-E).

(3) Normalise the tunel+ cells based on the number of KCs in PP vs PC regions.

After normalizing KCs death to KCs numbers in periportal (PP) versus pericentral (PC) regions, we found the proportion was significantly higher in PV regions compared to CV regions at 8 weeks of HFHC feeding. We have therefore revised our texts. (Revised manuscript, page 5, lines 194-201).

(4) Demonstrate the viability of KCs in vitro across conditions.

To confirm the identity and health of isolated KCs in our in vitro studies, we show that ~95% of primary isolated KCs are TIM4^+^ (Revised Figure S3A). Furthermore, Calcein-AM staining confirmed that the remaining KCs under our experimental conditions are viable and healthy (Revised Figure S4A).

(5) Confirm previous studies demonstrating different degrees of KC loss depending on the model of MASLD.

We thank the reviewer for highlighting this point. Consistent with previous studies, KCs loss has been reported to varying degrees depending on the MASLD model used, reflecting the heterogeneity of hepatic macrophages, marker choice, mouse husbandry, and diet regimen. For example, in a 6-week MCD feeding model, ~10% of CLEC4F^+^ KCs were TUNEL^+^ (Figure 4A, Ref. 9). Another 6-week MCD study reported a drop from 66% to 26% TIM4^+^ KCs (Figure 2A, Ref. 12). In an HFD model, TIM4^+^ KCs decreased by ~20% after 16 weeks (Figure 1G, Ref. 1). In a Western diet model, TIM4^+^KCs decreased by >50% at 36 weeks (Figures 1J and 2C, Ref. 2). Together, these studies underscore the model-dependent nature of KCs loss and highlight the importance of experimental context and marker selection when assessing KCs dynamics in MASLD. We have included these studies in our discussion section (Revised manuscript, page 9-10, line 393-402)

(6) Demonstrate in vivo that loss of CHIL1 drives further glycolysis in KCs.

In Figure 6G-H of our previous study, we showed that Chi3l1 deficiency leads to more glucose uptake by KCs *in vivo* whereas suppelementing KO mice with recombinant Chi3l1 will significantly reduced glucose uptake by KCs through treating mice with a fluorescent glucose analog 2-NBDG. We included the related figure here as Author response image 3.

**Author response image 3. sa3fig3:** Chi3l1 limits glucose uptake by Kupffer cells *in vivo*. (A) Measurement of 2-NBDG (a fluorescent glucose analog) uptake by KCs *in vivo*. WT and Chil1^-/-^ mice, either untreated or supplemented with rChi3l1, were injected intraperitoneally with 12 mg/kg 2-NBDG. After 45mins, KCs were isolated and glucose uptake assessed by spectrophotometry. (B) Representative immunofluorescence images of liver sections stained for TIM4 (red) and 2-NBDG uptake (green) to visualize glucose uptake by KCs in situ. Scale bar = 10 µm (zoom). Quantification is shown as the percentage of TIM4^+^ cells that are also 2-NBDG^+^. Representative images were shown in B. One-way ANOVA was performed in A, B. P value is as indicated.

(7) There is no mention of the publication of the metabolomics dataset; this should be released with the manuscript.

We included the raw metabolomics dataset as Table S1 and S2 now.

**Reviewer #3 (Recommendations for the authors):**
(1) Methods: Reconsider which methods are described in the main text versus the Supplementary Information to improve readability and consistency.

Thank you for your valuable suggestion. We have reevaluated and adjusted the placement of the methods section between the main text and the supplementary materials.

(2) Line 34: Check for grammar issues.

L34 has been revised as follows : Additionally, using Chi3l1-deficient mice, we further demonstrated that increased glucose utilization accelerates KCs death in vivo.

(3) Lines 101, 110: Explicitly reference the corresponding Supplementary Methods sections.

We have included the references for these two methods sections (Revised supplementary materials and methods, Line 30, 65, respectively).

(4) Figure 2: Iba1 marks all macrophages, not only monocyte-derived macrophages; both figure and text (line 205) require correction.

We have corrected Iba1 represent hepatic macrophages including both KCs and MoMFs (Revised Figure 2C, manuscript page 5, line 182).

(5) Line 218-219: Avoid overinterpretation, as only KCs, hepatocytes, and hepatic stellate cells were assessed - not all hepatic populations.

We appreciate the reviewer’s valuable suggestion and rephrased our description accordingly (Revised manuscript, page 5, line 186-189).

(6) Line 262: Use abbreviations consistently throughout the manuscript.

We have gone through the whole manuscript and double checked the abbreviations.

(7) Line 264: Include the palmitic acid (PA) concentration used.

We included 800 µM PA in the revised manuscript (Revised manuscript, page 6, line 250).”

(8) Lines 316-317: Check for grammar errors.

Grammar errors are checked (Revised manuscript, page 8, line 340-341).

(9) Line 337-338: See comment above on gating strategy.

We updated gating strategy accordingly (Revised manuscript, page 9, line 361-362).

(10) Line 343-344: Note that Chi3l1 is not exclusively expressed by KCs.

We rephrased our words accordingly (Revised manuscript, page 9, line 374-378).

(11) Lines 355-358: The statement that "sustained glycolytic hyperactivation culminates not in sustained activation, but in apoptotic cell death" is unsupported by data or literature, as macrophage polarization was not analyzed in this study.

We removed the statement from the revised manuscript.

(12) Lines 375-379: Rephrase to clarify that while KCs are metabolically active and glucose-demanding, excessive glycolytic flux accelerates apoptosis.

We have rephrased to clarify (Revised Manuscript, page 10, lines 405-407).

(13) Lines 375-385 & 387-397: Consolidate overlapping statements for conciseness and coherence.

We have consolidate the overlapping statements (Revised manuscript, page 10, lines 405-425).

Reference

Daemen, S. et al. Dynamic Shifts in the Composition of Resident and Recruited Macrophages Influence Tissue Remodeling in NASH. Cell Rep 34, 108626, doi:10.1016/j.celrep.2020.108626 (2021).

Remmerie, A. et al. Osteopontin Expression Identifies a Subset of Recruited Macrophages Distinct from Kupffer Cells in the Fatty Liver. Immunity 53, 641-657.e614, doi:10.1016/j.immuni.2020.08.004 (2020).

Ozer, J., Ratner, M., Shaw, M., Bailey, W. & Schomaker, S. The current state of serum biomarkers of hepatotoxicity. Toxicology 245, 194-205, doi:10.1016/j.tox.2007.11.021 (2008).

Malhi, H. & Gores, G. J. Molecular mechanisms of lipotoxicity in nonalcoholic fatty liver disease. Semin Liver Dis 28, 360-369, doi:10.1055/s-0028-1091980 (2008).

Ibrahim, S. H., Hirsova, P. & Gores, G. J. Non-alcoholic steatohepatitis pathogenesis: sublethal hepatocyte injury as a driver of liver inflammation. Gut 67, 963-972, doi:10.1136/gutjnl-2017-315691 (2018).

Kerr, J. F., Wyllie, A. H. & Currie, A. R. Apoptosis: a basic biological phenomenon with wide-ranging implications in tissue kinetics. British journal of cancer 26, 239-257, doi:10.1038/bjc.1972.33 (1972).

Poon, I. K., Lucas, C. D., Rossi, A. G. & Ravichandran, K. S. Apoptotic cell clearance: basic biology and therapeutic potential. Nat Rev Immunol 14, 166-180, doi:10.1038/nri3607 (2014).

Krenkel, O. & Tacke, F. Liver macrophages in tissue homeostasis and disease. Nat Rev Immunol 17, 306-321, doi:10.1038/nri.2017.11 (2017).

Tran, S. et al. Impaired Kupffer Cell Self-Renewal Alters the Liver Response to Lipid Overload during Non-alcoholic Steatohepatitis. Immunity 53, 627-640.e625, doi:10.1016/j.immuni.2020.06.003 (2020).

O'Neill, L. A. & Pearce, E. J. Immunometabolism governs dendritic cell and macrophage function. J Exp Med 213, 15-23, doi:10.1084/jem.20151570 (2016).

Vander Heiden, M. G. & DeBerardinis, R. J. Understanding the Intersections between Metabolism and Cancer Biology. Cell 168, 657-669, doi:10.1016/j.cell.2016.12.039 (2017).

Zhang J, Wang Y, Fan M, Guan Y, Zhang W, Huang F, Zhang Z, Li X, Yuan B, Liu W, Geng M, Li X, Xu J, Jiang C, Zhao W, Ye F, Zhu W, Meng L, Lu S, Holmdahl R. Reactive oxygen species regulation by NCF1 governs ferroptosis susceptibility of Kupffer cells to MASH. Cell Metab. 2024 Aug 6;36(8):1745-1763.e6. doi: 10.1016/j.cmet.2024.05.008. Epub 2024 Jun 7. PMID: 38851189.